# Cryo-electron tomography pipeline for plasma membranes

Willy W. Sun [1,3], Dennis J. Michalak [1,3], Kem A. Sochacki[1,3] ✉, Prasanthi Kunamaneni [1,2], Marco A. Alfonzo-Méndez[1], Andreas M. Arnold [1], Marie-Paule Strub[1], Jenny E. Hinshaw [2] ✉ & Justin W. Taraska [1] ✉

Cryo-electron tomography (cryoET) provides sub-nanometer protein structure within the dense cellular environment. Existing sample preparation methods are insufficient at accessing the plasma membrane and its associated proteins. Here, we present a correlative cryo-electron tomography pipeline optimally suited to image large ultra-thin areas of isolated basal and apical plasma membranes. The pipeline allows for angstrom-scale structure determination with subtomogram averaging and employs a genetically encodable rapid chemically-induced electron microscopy visible tag for marking specific proteins within the complex cellular environment. The pipeline provides efficient, distributable, low-cost sample preparation and enables targeted structural studies of identified proteins at the plasma membrane of mammalian cells.

Cryo-electron tomography (cryoET) offers a unique perspective of high-resolution protein structure within the crowded cellular environment. While cryoET in cells is powerful, it has two fundamental constraints. First, samples must be thinner than a typical eukaryotic cell[1,2]. Second, low signal-to-noise images and a crowded environment make it challenging to identify proteins smaller than 400 kDa[3], making it currently unsuitable for >99% of the human proteome[4]. These limitations and the need for increased efficiency and accessibility motivate the development of additional cell-thinning and protein-identification methods.

Cryo-focused ion or plasma beam (FIB)-milling has become a staple for cryoET by producing thin (<200 nm) samples rich in cellular structures from the heart of cells and tissues. FIB-milling is, however, time-consuming, expensive, complex, and not well-suited for the plasma membrane. Thus, other methods are needed. Cell unroofing, a fast and inexpensive technique classically used for platinum replica electron, atomic force, and fluorescence microscopies, isolates cellular plasma membranes with a thin associated protein cortex[5,6,7,8]. Yet, producing plasma membranes from cells on fragile cryo-EM grids is not trivial[9,10,11], and has not been used or characterized for quantitative structural cryoET.

Another common problem for cryoET is protein identification. To address this, EM-visible protein cages are an attractive, genetically encodable solution. Encapsulins (20–42 nm)[12–14] and iron-sequestering ferritins (12 nm)[15,16] are two examples of established electron microscopy tags with unique shapes. While these tags are large in comparison to the average protein or protein complex, rapamycin-induced linkages have been used to tether these oligomeric complexes to proteins inside living cells[14,16]. However, encapsulin cages can require 15 min to several hours to label their targets and are more sterically inhibiting than the smaller ferritin particles. The rapamycin-inducible fluorescent ferritin tagging system, FerriTag, is faster, binding to its target within seconds[16] but is not yet an established probe for cryoET. While previous thin-section transmission EM studies loaded FerriTag with iron to enhance contrast[16], metal contrast in cryoET can obscure nearby proteins of interest and perturb high-resolution structural information. Thus, an iron-loaded particle is not optimal.

Here, we present a comprehensive cell unroofing workflow that generates isolated basal or apical plasma membranes for correlative cryoET. We show that when cells are cultured on EM grids, membrane-associated protein complexes adapt to the surface topography of the grid. We demonstrate that cell unroofing preserves sub-nanometer

---

[1]National Heart, Lung, and Blood Institute, US National Institutes of Health, Bethesda, MD, USA. [2]National Institute of Diabetes and Digestive and Kidney Diseases, US National Institutes of Health, Bethesda, MD, USA. [3]These authors contributed equally: Willy W. Sun, Dennis J. Michalak, Kem A. Sochacki. ✉e-mail: kem.sochacki@nih.gov; jennyh@niddk.nih.gov; justin.taraska@nih.gov

resolution of ribosomes with subtomogram averaging. We use correlative light and electron microscopy (CLEM) and iron-free FerriTag as a chemically-inducible probe to locate and observe clathrin-associated proteins. In summary, we present and characterize an optimized pipeline for cryoET of plasma membranes capable of identifying specific proteins for structural cellular biology.

## Results

### Basal and apical plasma membranes can be isolated on grids

While cell unroofing has been an integral experimental technique for cell biology[5,7,8,10,17], the process itself and the parameters used have not been standardized for cryoEM grids. To standardize the cell unroofing process and to establish a general set of parameters, we assembled a simple air pressure-driven fluid delivery device (Figs. S1, S2) and tested how the syringe-to-sample distance and the delivery pressure affect cell unroofing. We found that a syringe fluid spray from a 20-gauge needle backed by 0.7–0.8 bar of air pressure at a distance of 1 cm effectively unroofs the cells without destroying Quantifoil cryoET grids (Fig. S3). These settings were used for isolating plasma membranes on grids.

Because of their different possible structures and functions, we sought to isolate basal and apical plasma membranes of cultured cells for cryoET. For basal membranes, cells were grown on Quantifoil cryoET grids. The grid was adhered to a coverslip using a PDMS (polydimethylsiloxane) stencil before cell plating. The stencil-grid-coverslip assembly could be handled as a single unit, minimizing direct grid manipulation during unroofing (Fig. 1a). After one day of culture, unroofing was performed with a stream of paraformaldehyde-containing buffer from a syringe as described above.

Apical plasma membrane preparations were performed by growing cells on a glass coverslip. To transfer the cells to an EM grid, the coverslip was rinsed in serum-free buffer and the carbon side of a poly-L-lysine-coated grid was lightly touched to the coverslip for 3–5 s and lifted (Fig. 1b). Cells transferred to the grid appeared largely intact as viewed by fluorescence, scanning electron microscopy, and FIB-milled cryoET (Fig. S4). The cells on the grid, now in an inverted orientation relative to their growth on the glass coverslips, were unroofed to generate apical plasma membranes on grids (Fig. 1b).

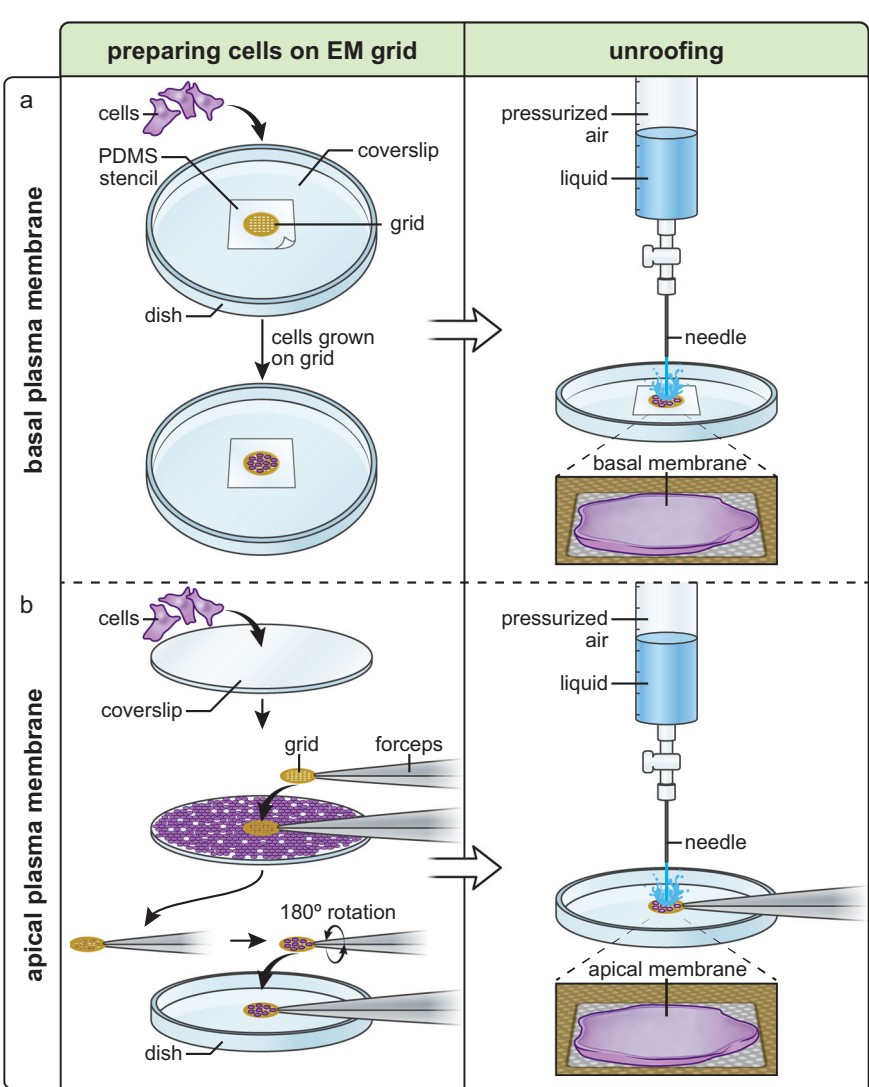

**Fig. 1 | Generating isolated plasma membranes on EM grids. a** Diagrams showing the workflow for isolating basal plasma membranes. A plasma-cleaned grid is placed onto a coverslip and secured with a PDMS stencil. Cells are then seeded onto the grid and incubated overnight. To generate isolated basal plasma membranes, the grid is placed under a pressurized fluid-delivering device and the grid is sprayed with unroofing buffer to wash away the apical portions of cells with a shearing force. **b** Diagrams showing the workflow for isolating apical plasma membranes. Cells are first seeded onto a coverslip and incubated overnight. A plasma-cleaned, poly-L-lysine coated grid is then brought into contact with the coverslip to pick up cells. Pressurized fluid is then applied to the grid to wash away the basal portions of cells to generate isolated apical plasma membranes.

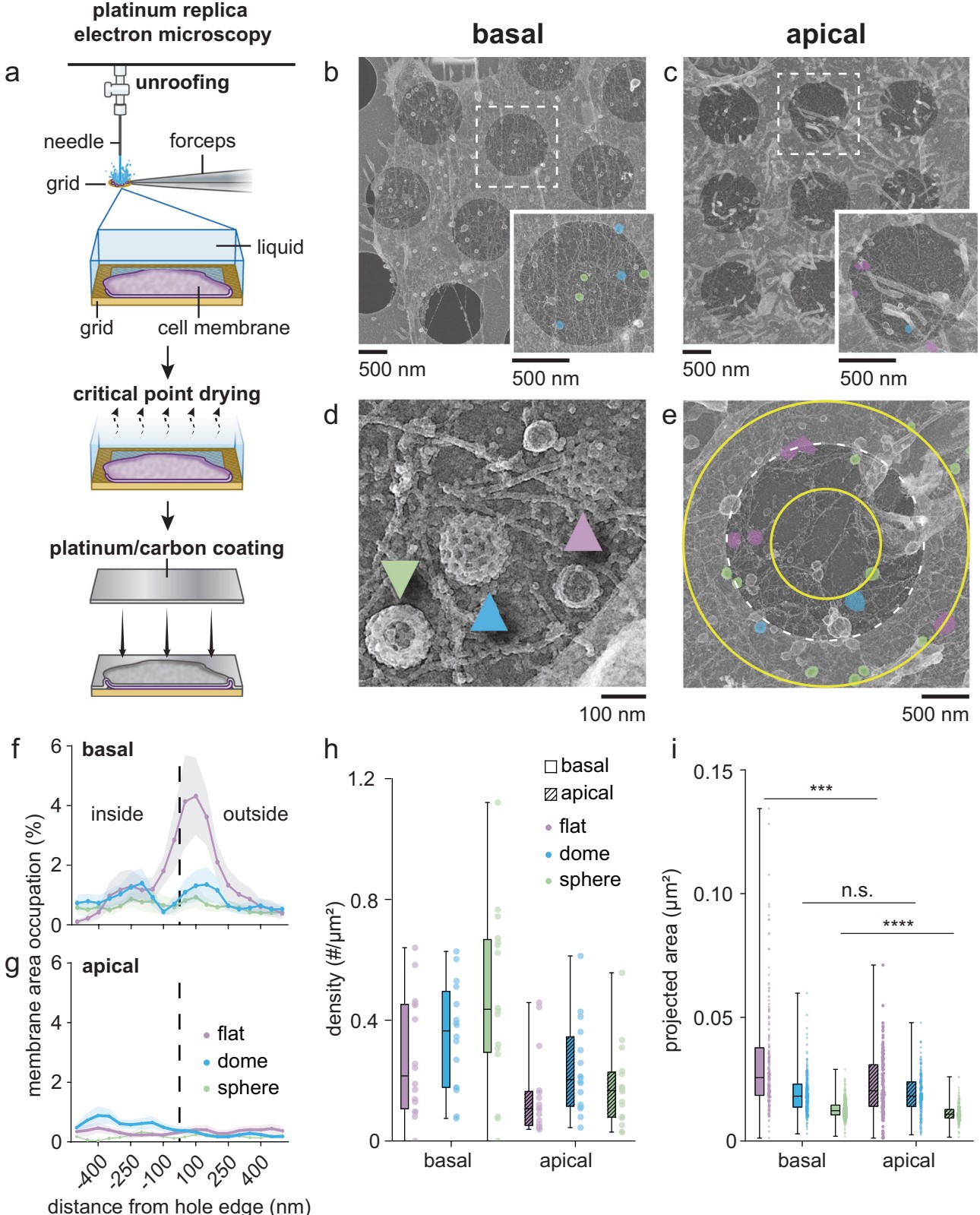

Isolated basal and apical plasma membranes could then be used for blotting, vitrification, and imaging.

## Grid topography influences organelle distribution on membranes grown on grids

To characterize differences in the organization of cellular structures on grid-bound basal and apical plasma membranes, we used platinum replica electron microscopy (PREM) of genome-edited HSC-3 (human squamous cell carcinoma) cells expressing endogenous EGFR-GFP (Fig. 2a–c)[18]. Both apical and basal membrane types showed a range of actin filament organization, vesicles, caveolae, and clathrin-coated structures (Figs. S5–S7). This cell type exhibits many filopodia which are prominent in apical membrane preparations. To quantify differences in the morphology of clathrin-coated structures within each

**Fig. 2 | Evaluating isolated plasma membranes of HSC-3 cells on EM grids with platinum replica electron microscopy. a** A cartoon showing the generation of platinum replicas of the isolated plasma membranes on grids. **b** An isolated HSC-3 cell basal plasma membrane on an R2/1 Quantifoil grid. The inset shows an enlarged view of the white dashed square area. Structural classes of clathrin are color-coded: lilac = flat; cyan = dome; green = sphere. **c** An isolated HSC-3 cell apical plasma membrane on an R2/1 Quantifoil grid. The inset shows an enlarged view of the white dashed square area. Color-coding is as in (**b**). **d** A close-up view of the isolated basal plasma membrane showing the three classes of clathrin structures. The lilac arrow points to a flat clathrin structure, the cyan arrow points to a dome-shaped clathrin structure, and the green arrow points to a spherical clathrin structure. **e** The edge of the hole (white dashed circle) is used as the reference point to evaluate the distribution of the different classes of clathrin-coated structures across the changing grid surface (1000 nm range, yellow circles inside and outside of the hole denote a 500 nm distance from the hole edge). **f, g** Comparison of the distribution of flat, dome, and sphere clathrin-coated structures (mean ± SEM) with respect to the edge of the hole between basal (**f**) and apical (**g**) isolated plasma membranes. **h, i** are box and whisker plots (box: 25th–75th percentile; whiskers: min to max) with individual data points shown to the right. **h** Comparing the density of different classes of clathrin structures between isolated basal and apical plasma membranes, and (**i**) of the projected area of individual clathrin structures grouped by structural class (Mann-Whitney test, two-tailed). ***$p = 0.0002$; ****$p < 0.0001$. For (**h, i**), horizontal lines = median. Images are representative of $N = 4$ grids (basal) and $N = 4$ grids (apical). For (**f–i**), $N = 16$ membranes, 203 flat, 237 domed, and 404 spherical clathrin structures from 2 grids (basal) and $N = 16$ membranes, 139 flat, 206 domed, and 173 spherical clathrin structures from 1 grid (apical).

preparation, we used semi-automated segmentation to identify three general classes: flat, dome, and spherical clathrin (Fig. 2d)[10]. The edge of the Quantifoil carbon film holes served as a reference to measure the distribution of clathrin-coated structures (Fig. 2e). Adherent basal plasma membranes exhibited more flat clathrin structures at or near the edge of the carbon film holes (Fig. 2f). Apical membranes exhibited no such preference (Fig. 2g). The addition of ultrathin carbon (2 nm) to the grids, which provides a carbon surface across the holes, lessened but did not eliminate this accumulation in basal membrane samples (Fig. S5). Overall, and as individual classes, the basal side contained more clathrin-coated structures (total projected area: 16.14 μm² from 844 structures; 749.46 μm² analyzed / # of structures per μm²: flat, $0.27 \pm 0.19$; dome, $0.35 \pm 0.17$; sphere, $0.47 \pm 0.29$) compared to the apical membrane (total projected area: 9.02 μm² from 518 structures; 1036.44 μm² analyzed / # of structures per μm²: flat, $0.15 \pm 0.13$; dome, $0.24 \pm 0.15$; sphere, $0.18 \pm 0.13$) (Fig. 2h). While the projected area of each class of clathrin structures varied (Fig. 2i), flat and sphere-shaped clathrin structures found in the basal membranes were larger (in μm²: flat, $0.032 \pm 0.022$; dome, $0.019 \pm 0.008$; sphere, $0.013 \pm 0.003$) than their apical counterparts (in μm²: flat, $0.023 \pm 0.012$; dome, $0.019 \pm 0.008$; sphere, $0.011 \pm 0.004$) (between flat structures: $p = 0.0002$; domes: n.s.; spheres: $p < 0.0001$). Both the apical and basal membranes from HSC-3 cells grown on coverslips have previously been evaluated with PREM[17]. Here, HSC-3 cells grown on grids exhibited a higher basal ($2.21 \pm 0.89\%$ grid vs. $0.84 \pm 0.1\%$ coverslip) and apical ($0.97 \pm 0.48\%$ grid and $0.56 \pm 0.17\%$ coverslip) clathrin-coated membrane area fraction than that previously reported for HSC-3 basal and apical membranes on coverslips. Many factors, including cell type influence clathrin size and density. For reference, HSC-3 cells exhibit a similar amount of clathrin on their membranes in comparison to many other previously reported cell types (HSC-3 on grids: $2.21 \pm 0.89\%$; HSC-3 on coverslips unstimulated: $0.84 \pm 0.1\%$; 3T3 on coverslips: $3.04 \pm 1.35\%$; BS-C-1: $1.47 \pm 0.8\%$; HeLa: $3.43 \pm 2.52\%$)[10,17]. Our data highlight that substrate topography affects clathrin assembly at the plasma membrane. Importantly, when cells are grown on grids with complex topographies, the grid surface represents a unique structural challenge, and cellular objects may exhibit distinct distributions across the sample. Understanding these differences is key for future work on the structure and function of organelles on these heterogeneous substrates.

## Unroofing provides thin membrane samples suitable for cryoET

After unroofing, basal and apical membrane preparations of HSC-3 cells were back-blotted, plunge-frozen, and observed with cryoET (Fig. 3a). With low or medium magnification imaging, membranes appeared light gray, only distinguishable by close inspection (Fig. 3b, c). Tomograms acquired from these membranes exhibited high contrast and were rich in membrane-bound organelles, actin, clathrin, intermediate filaments, and ribosomes (Fig. 3d, e). As sample thickness limits attainable resolution in cryoET[19,20], the average thickness of each tomogram was measured. Basal tomogram thickness

spanned from 78 to 196 nm and apical tomogram thickness spanned from 110 to 221 nm (Fig. 3f). The average thickness of isolated membrane tomograms used in this study was $163 \pm 52$ nm (mean ± stdv) (Fig. S8). Thus, unroofed samples have thicknesses rivaling the thinnest FIB-milled samples and provide high-contrast 3D views of a cellular environment.

## High-resolution structural information is preserved in unroofed cells

Ribosomes were dense and recognizable in our tomograms. They were observed densely on or near internal membranes and dispersed at or above the plasma membrane. To assess the preservation of high-resolution structural information in vitrified unroofed samples, we performed subtomogram averaging (STA) on ribosomes in an apical preparation of HEK293 cells (Fig. 4a; Table 1, grid 5). 111 tomograms were used to refine 11,249 ribosomes to obtain a consensus structure of the full 80S ribosome at 7.5 Å nominal resolution (Figs. 4b, c, S9). Classification without alignment was performed and revealed two classes which resembled a pre-translocational rotated state and a non-rotated state (Figs. 4d, S9)[21–23]. The rotated state exhibited tRNA in the hybrid A/P, P/E sites (Fig. 4d, blue and pink respectively). The non-rotated state exhibited prominent density in the P state (Fig. 4d, orange). Separate focused classification at the peptide exit site revealed a coarse structure of a putative translocon and confirmed the presence of plasma membrane proximal endoplasmic reticulum in many of our tomograms (Figs. 4e, S10). Together, these data show that particles found within unroofed membrane samples retain sub-nanometer structural information.

The air-water interface (AWI) is known to disrupt protein structures observed in purified single particles averaging cryo-samples[24]. Similarly, vitrified unroofed cells are bounded by two AWIs either proximal to the extracellular side of the plasma membrane (bottom) or distal to the intracellular side of the plasma membrane (top) (Fig. 4f). To examine how the AWI affected ribosome structure in unroofed samples, the AWI-particle distances were calculated from 111 tomograms (Fig. 4g). The complete set of segmented particles (Fig. 4g, purple) was compared to the particles that grouped into well-aligned classes and therefore contributed to the final consensus map (Fig. 4g, cyan). A peak in the particle distribution, indicating accumulation, occurred at 25–30 nm from the top AWI. In contrast, the distribution measured from the bottom AWI does not show accumulation but reaches a plateau in the number of particles 35–40 nm from the bottom AWI (Fig. 4g, left). The lack of accumulation is likely due to the plasma membrane acting as a barrier between the AWI and intracellular material and the extra distance accounts for the thickness of the plasma membrane. The proportion of particles in the well-aligned class was used as an indicator of particle quality (Fig. 4h). The percentage of particles selected for subsequent processing was mostly constant throughout the portion of the tomogram containing biological material even in the region of accumulated ribosomes near the top AWI. We conclude that particle quality is retained in the regions more than

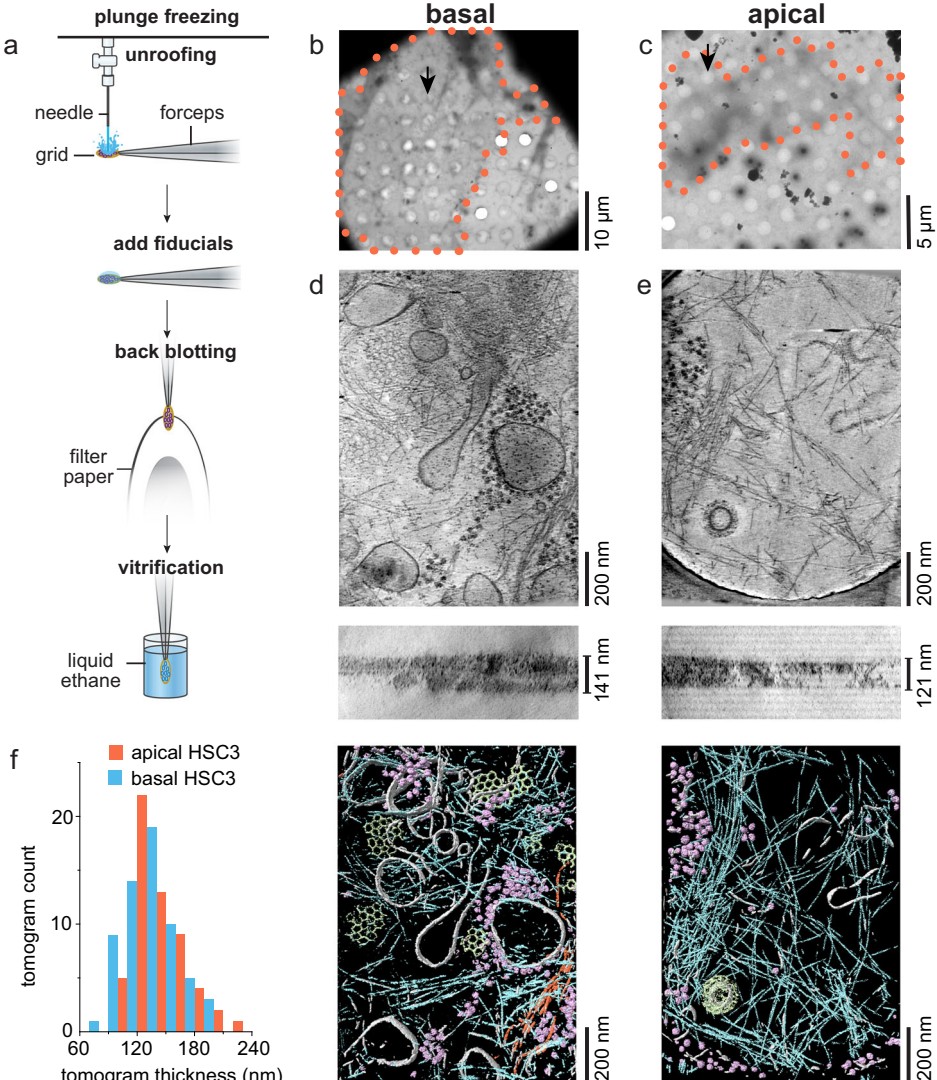

**Fig. 3 | Unroofed cells provide 100–200 nm thick plasma membrane samples for cryoET. a** A diagram highlighting unroofing, addition of fiducial markers, back-blotting, and plunging in liquid ethane for vitrification. **b** A 2250x magnification montage of a grid square containing an isolated HSC-3 basal membrane (outlined with an orange dotted line). **c** A 2250x magnification montage of a grid square containing an isolated apical HSC-3 membrane (outlined with an orange dotted line). **d** (top) shows a minimum-intensity projection along the Z-axis through 21 slices of a Gaussian-smoothed bin-8 tomogram acquired at the location of the arrow in (**b**). **d** (middle) shows a minimum-intensity projection of 101 slices through the Y-axis of the same tomogram with the measured thickness shown. **d** (bottom) shows a segmentation (mask-guided isosurface) of the above tomogram: gray = membrane, purple = ribosomes, blue = actin, light green = clathrin, orange=intermediate filaments. **e** Same as (**d**), but for the apical membrane tomogram acquired at the position of the black arrow in (**c**). **f** Histogram of tomogram thicknesses of HSC-3 basal and apical membranes imaged here. $N_{apical}$ = 56, $N_{basal}$ = 61, 2 grids represented for each.

25 nm from the AWIs (dashed lines), where ribosomes are identified. However, acute damage of particles within 25 nm from the top AWI is assumed to be possible and likely.

**Correlative imaging facilitates tomogram acquisition of stalled endocytic events**

To locate rare proteins or proteins of unknown structure, we implemented a CLEM protocol. The HEK293 cell line described above expresses, upon induction, a fluorescently labeled well-characterized dominant negative mutant of dynamin 1, Dyn1(K44A)-GFP, that blocks clathrin-mediated endocytosis[25–27]. Using this cell line, we tested our CLEM protocol to search for sites of arrested clathrin-mediated endocytosis and dynamin accumulation. After apical unroofing, we added 500 nm red fluorescent poly-styrene beads which adhered to the grid, specifically in poly-L-lysine rich locations lacking plasma membrane. This image registration allowed us to precisely identify

grid holes containing GFP fluorescence for tomogram acquisition (Fig. 5a–e). We acquired 198 tomograms on two grids and identified N = 490 clathrin structures within the tomograms. We observed N = 42 dynamin decorated tubules (9% of clathrin structures) where dynamin could be easily identified based on its characteristic polymer spiral (Fig. 5f)[25]. This is similar to the 10% previously reported in HeLa cells in resin sections[28]. Conversely, we commonly saw large arrested clathrin structures (N = 219, 45% of clathrin structures) that resembled clathrin grape clusters (Fig. 5g)[29].

Though there was fluorescence at these sites, dynamin in clathrin grape clusters was more difficult to identify than when it assembles in tubes, presumably because it was assembled in short spirals or oligomers that were not readily distinguishable. The difficulty in identifying dynamin in each of these tomograms highlighted the need for a more precise EM-visible protein tag. Yet, the high density of arrested clathrin at these sites confirmed our ability to use

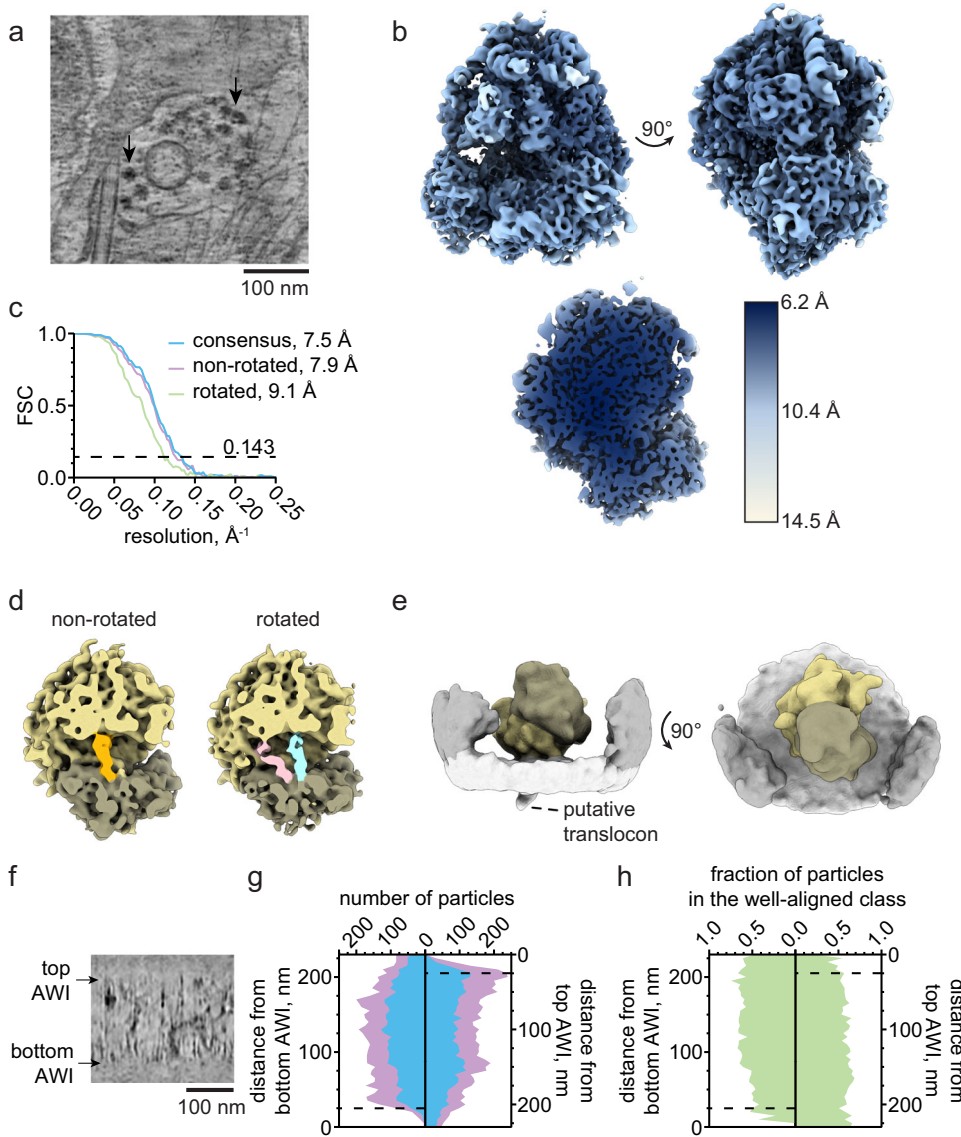

**Fig. 4 | Subtomogram averaging and contextual analysis show sub-nanometer detail is preserved in isolated plasma membranes. a** Projection of 21 z-slices from a tomogram of an isolated plasma membrane representative of the 111 tomograms used in this analysis. 80S ribosomes (black arrows) are frequently found in unroofed HEK293 cells overexpressing dynamin-1(K44A). **b** Rotated views (top) and a clipped view (bottom) of a consensus subtomogram average filtered according to the local resolution. **c** Fourier shell correlation (FSC) profiles obtained from subtomogram averages. The nominal resolution is reported at FSC = 0.143. **d**, **e** Classification of the set of well-aligning particles obtained subtomogram averages of the 80S ribosome in non-rotated and rotated states. In (**d**), tRNA occupying the P, P/E, and A/P sites are indicated in orange, pink, and blue,

respectively. A subset of 446 membrane-bound ribosomes is shown (two views, (**e**)). **f** A view from a tomogram with the top and bottom air-water interfaces (AWIs) indicated (black arrows). **g** Distances of putative ribosomes were measured from both AWIs. The number of particles within successive 5 nm bins from the bottom and top AWIs (left and right panels, respectively) is plotted for the set of particles obtained immediately following picking (cyan) and the set of well-aligning particles obtained from classification (purple). **h** The fraction of particles found with particle picking that constitute the well-aligning class are plotted with respect to their distance to the bottom and top AWIs. Dashed lines in (**g**, **h**) indicate 25 nm distances from each AWI. This condition is represented in grids 5-6, all ribosome data are from grid 5 (Table 1).

CLEM to find sites where fluorescently labeled proteins were present. Further, we could utilize the high density of clathrin at these sites for subtomogram averaging of clathrin assembled at the plasma membrane. During assembly at the plasma membrane, flexible clathrin triskelia assemble into polyhedral lattices of varying shapes, sizes, and curvatures[10,30]. Previous high-resolution averaging has relied on assembly of reconstituted pieces into highly curved and more rigid clathrin baskets, vesicles, or purified clathrin vesicles[31–34]. In our data, clathrin is still bound to the plasma membrane in a state prior to endocytosis. Subtomogram averaging of clathrin vertices (Fig. 5h; an average of 1550 vertices from a subset of 20 tomograms, EMD-46973) results in a density similar to previous models. A rigid body fit of a

clathrin vertex from reconstituted clathrin and AP2 coated vesicles (PDB ID:6YAI)[32] shows structural conservation in the proximal region of the clathrin heavy chain and light chain (Fig. 5h, bottom left). The N-terminus of heavy chain in our density is splayed away from that of the reconstituted vesicle map (Fig. 5h, bottom center). We attribute this structural difference to the lattice accommodating a lower curvature on average when assembled on the plasma membrane prior to endocytosis. Additionally, we observed density neighboring the vertex consistent with the reported AP2 β2 appendage (Fig. 5h, bottom right). Superimposing this reconstruction onto subtomogram positions recreates the lattice organization of clathrin coated pits (Fig. 5i). Using a single consensus model was sufficient to

**Table 1 | CryoET Sample summary**

| | Cell line | Gene introduction | Specific treatment | Unroofing method | CLEM imaged | Quantifoil Grid used | Scope[a] | Fig. |
|---|---|---|---|---|---|---|---|---|
| Grid #1 | HSC-3 | | serum starved | basal | no | Au 300 R2/2, 2 nm carbon | Krios 3 | Figs. 3b, d, f S8 |
| Grid #2 | HSC-3 | | serum starved | basal | no | Au 300 R2/2, 2 nm carbon | Krios 1 | Figs. 3f S8 |
| Grid #3 | HSC-3 | | | apical | yes | Au 300 R1.2/1.3 | Krios 1 | Figs. 3c, e, f, S8 |
| Grid #4 | HSC-3 | | | apical | yes | Au 300 R1.2/1.3 | Krios 2 | Figs. 3f, S8 |
| Grid #5 | HEK293 Trex | Stably transformed with Dynamin1(K44A)-GFP | overnight doxycycline | apical | yes | Cu 200 R2/2 | Krios 3 | Figs. 4,5, S8–S10 |
| Grid #6 | HEK293 Trex | Stably transformed with Dynamin1(K44A)-GFP | overnight doxycycline | apical | yes | Au 300 R1.2/1.3 | Krios 2 | Fig. 5 Fig. S8–S9 |
| Grid #7 | HEK293 | Transient transfection GFP-FKBP-LCa/FerriTag | 2 min rapamycin | apical | yes | Au 300 R1.2/1.3 | Krios 1 | Figs. 6e, f, S8 |
| Grid #8 | HEK293 | Transient transfection GFP-FKBP-LCa/FerriTag | 2 min rapamycin | apical | yes | Au 300 R1.2/1.3 | Krios 1 | Figs. 6f, S8 |
| Grid #9 | HEK293 | Transient transfection Hip1R-GFP-FKBP/FerriTag | 2 min rapamycin | apical | yes | Au 300 R1.2/1.3 | Krios 1 | Figs. 6b-d, f-h, S8 |
| Grid #10 | HEK293 | Transient transfection Hip1R-GFP-FKBP /FerriTag | 5–6 min rapamycin | apical | yes | Au 300 R1.2/1.3 | Krios 1 | Fig. S8 |

*Krios 1*: Thermo Fisher Scientific Titan Krios G3 transmission electron microscope with xFEG operated at 300 kV, equipped with a Gatan Quantum LS imaging filter at 20 eV slit width and post-filter Gatan K3 direct electron detector.
*Krios 2*: Thermo Fisher Scientific Titan Krios G4 transmission electron microscope with xFEG operated at 300 kV equipped with a Gatan BioContinuum K3 detector at 20 eV slit width.
*Krios 3*: Thermo Fisher Scientific Titan Krios G1 transmission electron microscope with xFEG operated at 300 kV, equipped with a Gatan Bioquantum K3 detector at 20 eV slit width.
[a]Scopes (Microscopes used for final data acquisition).

visualize the polyhedral lattice construction including hexagonal, pentagonal, and heptagonal arrangements of triskelia.

## Iron-free ferritin tagging identifies Hip1R in cryo-electron tomograms of unroofed cells

To identify specific proteins within a tomogram, we tested the ability of iron-free rapamycin-induced FerriTag[16] to label Hip1R, a 119 kDa, 50–60 nm long clathrin adapter that makes a coiled-coil parallel homodimer linking the membrane to actin[35–39]. First, FerriTag recruitment to Hip1R was confirmed using fluorescence microscopy. Before rapamycin, HEK293 cells expressing Hip1R-GFP-FKBP and FerriTag (labeled ferritin heavy chain, FRB-mCherry-FTH1; and ferritin light chain, FTL) exhibited diffuse red cytoplasmic fluorescence and green fluorescent puncta, typical of clathrin-associated proteins, in total internal reflection fluorescence microscopy (TIRF). After 200 nM rapamycin addition, the red fluorescence colocalizes with the green Hip1R puncta (Fig. 6a; FerriTag, magenta; Hip1R, green). This redistribution was rapid and visible within 30 s.

Next, the Hip1R-FerriTag system was used to examine empty ferritin cages in cryoET. After 2-minute incubation in 200 nM rapamycin, we unroofed cells and observed the apical membranes in cryofluorescence and cryoET. At sites with both GFP and mCherry fluorescence, tomograms exhibited distinct 12-nm hollow spheres surrounding clathrin lattices (Fig. 6b–e). These spheres were prominent and could be automatically identified using a trained convolutional neural network (EMAN2). Distributions of Hip1R/FerriTag (as in Fig. 6d, Supplementary Movie 1) were compared to that of FerriTag coupled to the N-terminus of clathrin light chain A (GFP-FKBP-LCa; as in Fig. 6e, Supplementary Movie 2), a subunit of the heterohexamer triskelia that sits at the membrane-distal portion of the ~25 nm thick clathrin membrane coat. For both Hip1R and clathrin, the data exhibit a large peak within 100 nm of the clathrin-coated membrane with very low background labeling (clathrin, grid 7-8, 91 tomograms, 81% within 100 nm, $N_{<100} = 633$, $N_{tot} = 784$; Hip1R, grid 9, 68 tomograms, 81% within 100 nm, $N_{<100} = 1638$, $N_{tot} = 2057$)(Fig. 6f). Of the data within 100 nm, FerriTag is found $35 \pm 15$ nm away from the membrane for GFP-FKBP-LCa and is $50 \pm 17$ nm away for Hip1R. The Hip1R peak

plateaus between 38 and 52 nm, consistent with Hip1R radially projecting away from the clathrin lattice, or tilted at an angle (Fig. 6f, model). In its fully extended state, Hip1R has been shown to consist of a 40-nm long parallel coiled-coil dimer with two distinct 7–10 nm globular domains at each end[35]. Segmentation and enlarged tomogram slices show that several Hip1R can bind to a single actin fiber. Putative Hip1R densities are observed in extended and angled states and match the in vitro structure (Fig. 6g, h, Supplementary Movie 3–6). Together, these data demonstrate that iron-free ferritin is an efficient, expressible, chemically-induced EM and light visible probe for cryoET in unroofed samples to mark individual proteins.

## Discussion

We present a correlative cryoET pipeline for imaging identified proteins at the plasma membrane. An air-pressure-driven syringe unroofing technique provides a quick and simple method for preparing isolated grid-bound apical or basal membranes from mammalian cells. With PREM, we show that growing cells on a Quantifoil grid alters the organization of the basal membrane cortex, specifically quantified for clathrin. The thickness and content of the material in vitrified unroofed samples were found suitable for obtaining sub-nm protein structure with subtomogram averaging. Correlative cryo-fluorescence and iron-free FerriTag complement cell unroofing to identify membrane-associated proteins of interest. These methods facilitate visual proteomics of the eukaryotic cell plasma membrane.

Our PREM analysis of grid-bound membranes was performed on a single human cell line (HSC-3). It is expected that contents between the apical and basal membranes will differ in a cell-type dependent manner. However, evidence supports the idea that flat clathrin will likely concentrate at the edge of carbon holes in basal membranes (Fig. 2f) in most adherent cell types. Specifically, clathrin has been reported to concentrate over sites of high substrate curvature in MDA-MB-231 breast cancer cells[40]. Our observation of flat clathrin lattices at edges is consistent with the longer average clathrin-mediated endocytosis (CME) lifetimes observed at high substrate curvature[40]. Flat clathrin lattices in HSC-3 cells, rat myotubes, and many other cell lines have been directly linked to substrate adhesion with integrins[17,41,42].

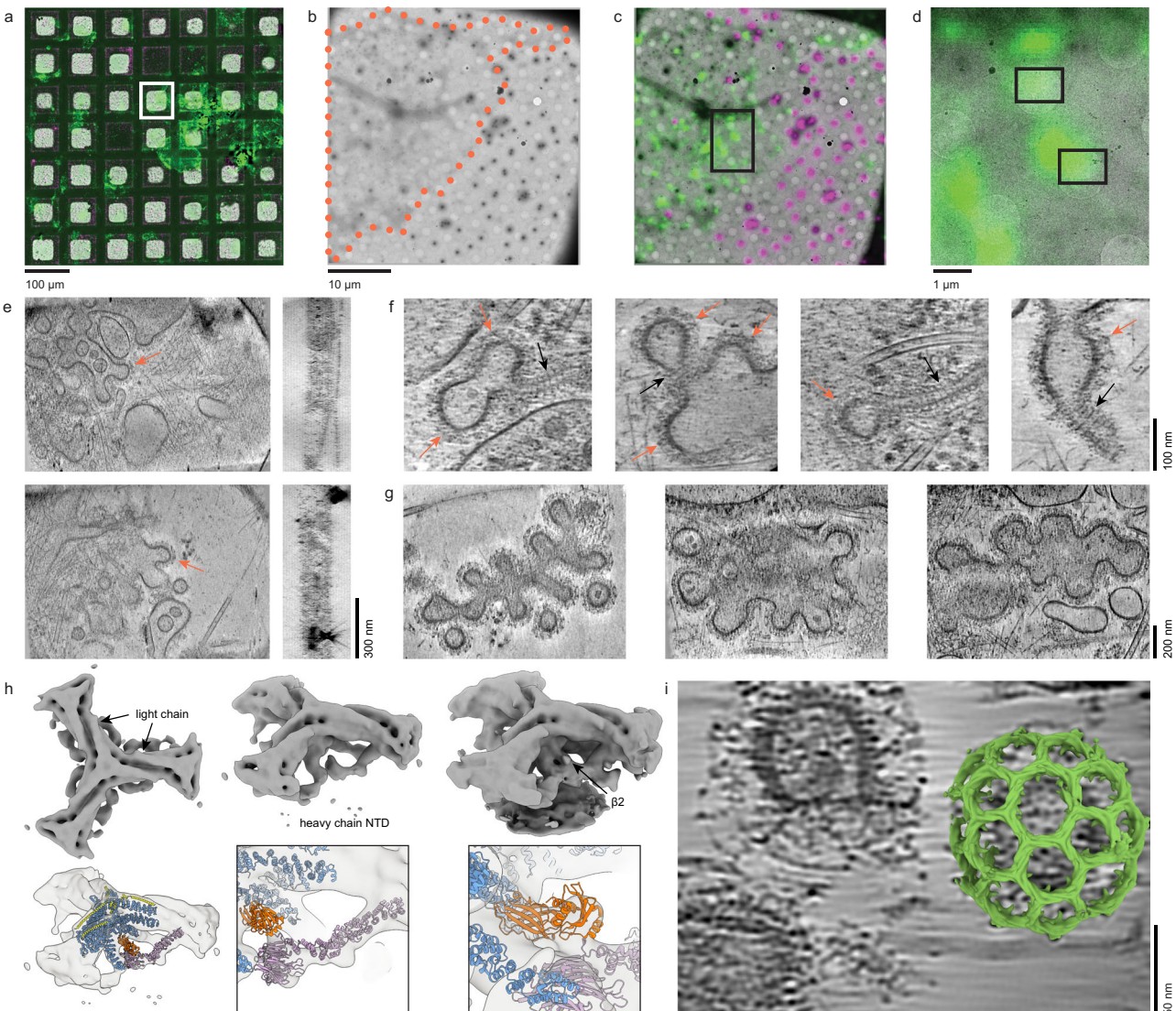

**Fig. 5 | CLEM finds sites of arrested clathrin-mediated endocytosis (CME).**
**a** Select portion of grid shown as a low magnification cryoEM image registered with cryo-fluorescence images of Dyn1(K44A)-GFP (green) and 500 nm fiducial markers (red). **b** The grid square highlighted in (**a**) (white square) is shown in a higher resolution map. Orange dots indicate the outline of the isolated plasma membrane. **c** The fluorescence overlay is shown. **d** The black box from (**c**) is shown enlarged. Black boxes indicate the location of tilt series acquisition for the tomograms shown in (**e**). **e** Examples of tomograms in XY (left) and XZ (right). **f** Examples of arrested CME sites with Dynamin 1 (K44A) tubules. In (**e**, **f**), orange and black arrows point to putative clathrin and dynamin densities, respectively. **g** Examples of arrested CME sites with large clathrin-decorated clusters. XY tomogram images in (**e**–**g**) are minimum intensity projections (mIPs) over 21 Gaussian-smoothed XY slices while XZ images are mIPs of 101 XZ slices. **h** A subtomogram average of clathrin (EMD-

46973) is shown with two representative thresholds (left and middle columns, 0.14; right column, 0.08). Labels indicate putative proteins and domains contributing to observed densities. A rigid body fit of PDB ID:6YAI is shown (bottom row) within the subtomogram average showing the clathrin heavy chain domains proximal to the central vertex (blue), clathrin light chain (yellow), clathrin heavy chain N-terminal domain with distal leg (pink), and the β2 appendage of the adapter AP2 (orange). Magnified views of the heavy chain N-terminal domain (bottom-middle) and the β2 appendage are shown (bottom-right). **i** The clathrin vertex reconstruction is superimposed onto refined subtomogram positions reconstructing a clathrin coated pit (green) representative of those found in the 20 tomograms used in this analysis. Several lattice arrangements are visible within the single structure. The image is an average of 10 XY slices from a denoised tomogram. This condition is represented in grids 5-6 (Table 1). All clathrin data are from grid 6.

---

Similarly, adhesive clathrin structures aggregate at high-curvature collagen-pinching membrane ridges during cell migration[43]. Thus, flat clathrin structures, observed in many cell types[10], are expected to exhibit adherent patches at the edge of carbon holes.

The air-water interface (AWI) is well-known to cause protein adsorption, preferred orientation, and damage in single particle cryoEM[44]. Here, unroofing exposes cellular material to an AWI. Notably, membrane-bound protein complexes are blocked from the bottom AWI by the membrane and most are kept from the top AWI due to their physical association with the plasma membrane. Ribosome positions in our data confirm that proteins can accumulate at the top AWI as the buffer retreats during blotting (Fig. 4g). While the STA

performed here on ribosomes achieves similar resolution to other FIB-milled ribosomal RELION 4.0-based STA[45], higher resolution has been obtained with FIB-milled samples with other STA software[22,46,47]. However, tomograms of unroofed cells exhibit particular advantages for STA efforts because decreased cytosolic crowding makes protein shape and position more distinct and, coupled with the lack of FIB radiation damage[48], enhances the potential for high-resolution STA.

Sites of arrested clathrin-mediated endocytosis were observed to contain large clathrin-coated superstructures (Fig. 5e–g). Our subtomogram average of clathrin lattice vertices at these sites resolves several structural features previously described for in vitro systems[31–34]. However, our structure highlights a displacement of the

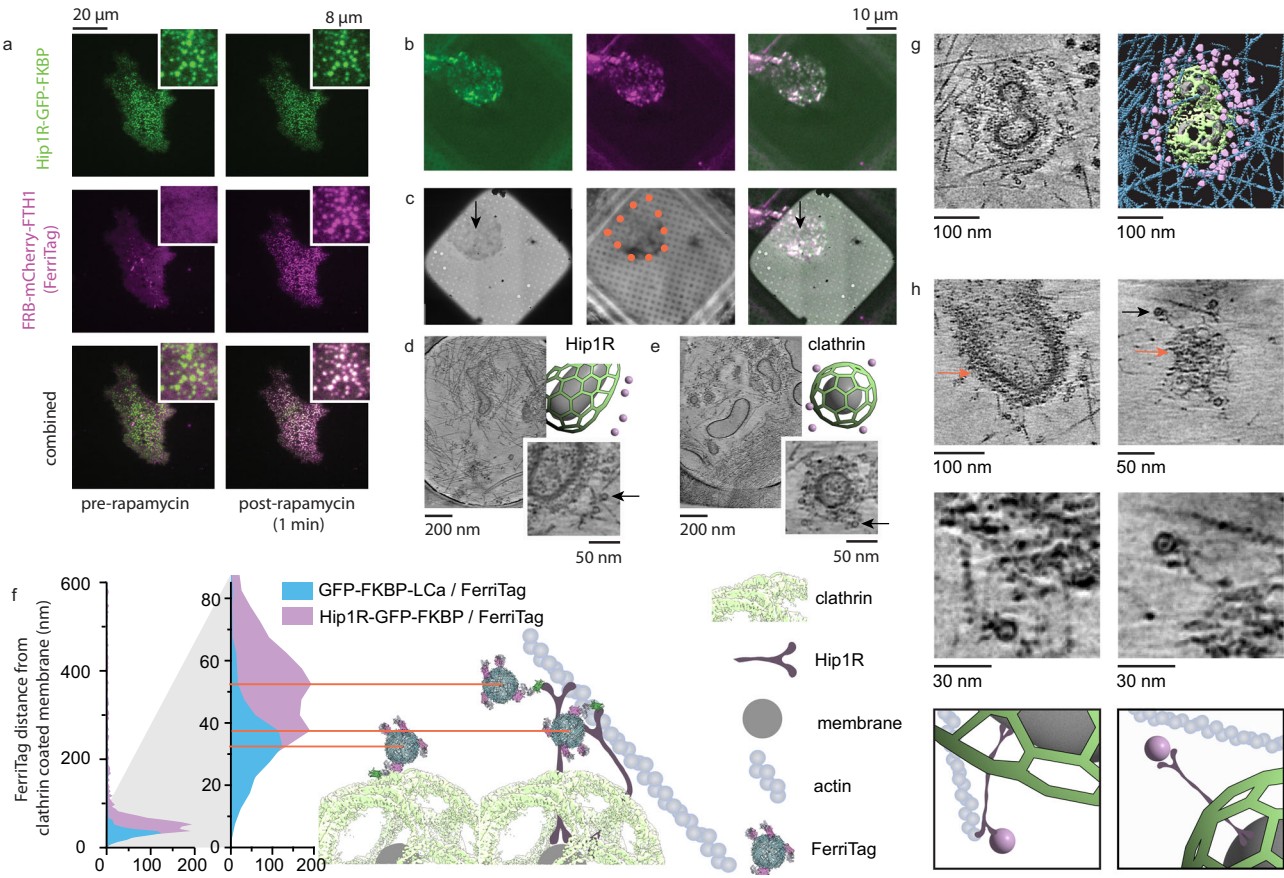

**Fig. 6 | Iron-Free FerriTag is specific, efficient, and visible in cryoET. a** TIRF microscopy of a HEK293 cell expressing FerriTag (FRB-mCherry-FTH1, magenta, and FTL) and Hip1R-GFP-FKBP (green) shown before (left) and 30-60 seconds after (right) rapamycin addition. Representative of 2 coverslips, 15 regions, 24 cells. **b** Cryo-fluorescence microscopy of HEK293 isolated plasma membrane on a grid expressing FerriTag (FRB-mCherry-FTH1, magenta; and FTL) and Hip1R-GFP-FKBP (green) with combined colors on the right. **c** The same grid in low-magnification CryoEM (left), reflection image used for fluorescence registration (middle), and combined EM and fluorescence image (right). Orange dots = membrane outline, arrows = position of (**d**) acquisition. **d** Tomogram of Hip1R/FerriTag labeling (right). Empty FerriTag structures are visible as 12 nm circles (arrow) (Supplementary Movie 1). **e** Selected tomogram from HEK293 cells with labeled FerriTag on GFP-FKBP-LCa (right)(Supplementary Movie 2). The cartoons depict membrane (gray), clathrin (green), and FerriTag (purple). **f** A histogram (5-nm bins) of the FerriTag distance from clathrin-coated membrane (shown at 0–600 nm and 0–80 nm) for clathrin light chain (GFP-FKBP-LCa, grid 7-8, $N$ = 784 FerriTags) and Hip1R (Hip1R-GFP-FKBP, grid 9, $N$ = 2057 FerriTags). Molecular models of Ferritin (PDB-1fha[68]), mCherry (PDB-2h5q[69]), FRB/FKBP (PDB-3fap[70]), GFP (PDB-5wwk[71]), and EM density of the clathrin cage (emdb-21608[34]) at scale with the zoomed-in histogram with cartoons of Hip1R, actin, and membrane as putative models. **g** An example of FerriTag Hip1R labeling around a clathrin structure (Supplementary Movie 3). Segmentation of membrane (gray), clathrin (green), FerriTag (purple), and actin (blue) are shown to the right. **h** Two examples (side-by-side) showing Hip1R density adjacent to FerriTag (Supplementary Movie 4-5). The top image displays the clathrin structure with clathrin (orange arrow) and a FerriTag (black arrow) highlighted and enlarged below with cartoons of membrane (light-gray), clathrin (green), Hip1R (dark gray), FerriTag (purple), and actin (blue) shown to aid image interpretation. Tomogram images are minimum intensity projections (mIPs) of 21 (**d**, **e**), 10 (**g**), 20 (**h**) Gaussian-smoothed XY-slices.

clathrin heavy chain N-terminus. We attribute this displacement to the lower average curvature of the clathrin lattice at the plasma membrane compared to purified or reconstituted vesicles. Alternate splaying of the N-terminus at different curvatures may help regulate adapter binding during different stages of endocytosis. Indeed, the major clathrin adapter AP2 binds in this region and we resolve density consistent with its β2 appendage[32]. This work emphasizes the importance of resolving adapter/heavy chain contacts during endocytic transitions at the plasma membrane rather than relying only on high curvature consensus structures from purified components or organelles. The methods described here have the potential to delve into this heterogeneity with higher resolution.

To identify specific proteins in the tomograms, we used empty FerriTag to label both Hip1R and clathrin light chain. Clathrin light chain, in particular, is an integral component of a dynamic and interlaced protein cage where bulky tags could be problematic. Still, the brief chemically-induced tagging required for FerriTag facilitates close

labeling while allowing for an intact (Fig. 6e) and functional lattice[16]. We have previously visualized the distribution of Hip1R around clathrin structures in HeLa cells using correlative super-resolution localization fluorescence microscopy and PREM[30]. In those lower-resolution 2D CLEM data, the fluorescence Hip1R signal uniformly covered flat clathrin structures but was enriched on the outside of curved clathrin structures. This was interpreted as the N-terminus splaying radially outward in curved structures. Here, cryoET of FerriTag-labeled Hip1R allows us to find the 3D position of individual Hip1R N-termini with respect to both clathrin and actin. This facilitates 3D quantification of membrane/Hip1R N-terminus distance for each FerriTag. When Hip1R/FerriTag was previously visualized in resin sections[16], the average distance of FerriTag from the coated membrane was 30 nm, 20 nm smaller than what was measured here. Conceivably, 70 nm sections result in a 2D projection of Hip1R that is angled toward or away from the plane of reference, making the distances appear shorter. Thus, the 3D measurement provides a histogram of

membrane/Hip1R N-terminus distances more consistent with the estimated length of Hip1R (50–60 nm). Hip1R/actin binding has previously been shown to be regulated by clathrin light chain[36]. The model postulates that Hip1R present throughout the clathrin lattice would not bind actin as readily as the Hip1R outside of the clathrin lattice at the budding membrane neck and may bend or fold inward to inhibit actin binding. Here, we find a dense network of actin and Hip1R-bound FerriTag uniformly surrounding the clathrin structures consistent with Hip1R actively binding actin throughout the entire clathrin lattice. While our data are consistent with Hip1R bending, it is in close proximity to actin. FerriTags line up along actin fibers (Fig. 6g) emphasizing the need for some requisite Hip1R bending to connect a linear fiber to a curved clathrin structure. A recent cryoET study of CME actin forces showed putative Hip1R densities in the intact human skin melanoma cell line, SK-MEL-2[49]. These ~50 nm-long densities contain two arms at either end, cross the clathrin lattice, and are consistent with what we observed here. However, FerriTag is integral in confidently identifying these long, thin, and relatively dim densities as Hip1R, showcasing the utility of the EM techniques presented here.

The potential of cryoET and cryo-fluorescence is growing, and its impact on cell biology and medicine will depend on wider access. Greater accessibility requires more rapid and available sample preparation techniques. Here, we show that the preparation of vitrified unroofed cells is a powerful addition to the fast-evolving cryoET toolbox. These methods enable imaging of the many important plasma membrane proteins involved in the function of cells and diseases.

## Methods
### CryoET sample summary
Please see Table 1 for sample summary.

### Cell lines
The human (male, Japanese) tongue squamous cell carcinoma cell line, HSC-3, used here, is genome edited to express epidermal growth factor receptor (EGFR)-GFP (at 40-50% of EGFR) and was a kind gift from Dr. Alexander Sorkin[18]. It was maintained in DMEM (Thermo-Fisher 11995065) supplemented with 10% FBS, with or without 50 mg/mL streptomycin−50 U/mL penicillin (Thermo-Fisher 15070063). HEK293, human (female, unknown ethnicity) embryonic kidney, WT cells were purchased from ATCC (CRL-3216) and maintained in Eagle's minimum essential medium (EMEM, Fisher Scientific 502382632) supplemented with 10% fetal bovine serum (FBS). For FerriTag experiments, HEK293 cells were transfected with *FTL* (Ferritin light chain, Addgene plasmid # 100750), *FRB-mCherry-FTH1* (Ferritin Heavy Chain, Addgene Plasmid #100749), and one of either *HIP1R-GFP-FKBP* (Addgene Plasmid #100752) or *GFP-FKBP-LCa* (Addgene Plasmid #59353)[16] using the Lonza Nucleofector and kit V (Lonza #VCA-1003) immediately prior to coverslip seeding (western blot, Fig. S11a, b). 200 nM rapamycin was added to the cells for the indicated times to induce FKBP/FRB heterodimerization. The inducible HEK293 Dynamin1(K44A)-GFP cell line was generated as follows. The cDNA coding for human Dynamin1(K44A)-GFP (Addgene #34681) was inserted into *pcDNA5-FRT/TO* to generate *pcDNA/TO/Dyn1-K44A-GFP* using the In-Fusion cloning kit (Takara), and the following primers: ATGGTGAGCAAGGGCGAGG, ACGCTAGAGTCCGGAGGC, TCCGGACTCTAGCGTCCTGCCATGGG-CAACCGC, GCCCTTGCTCACCATGGTGGCGACCGGTGGATCC. Proper insertion was confirmed by full sequencing (Plasmidsaurus). To generate inducible cells expressing Dynamin1(K44A)-GFP, parental Flp-In T-Rex HEK293 cells (Thermo Fisher, R78007) were co-transfected with a 1:5 mass ratio of *pcDNA/TO/Dyn1-K44A-GFP* and the Flp recombinase expression plasmid, *pOG44* (Thermo Fisher, V600520) using Lipofectamine 3000 (Invitrogen, L3000-015) following manufacturer's instructions. 24 h after transfection, cells were subjected to selection with 5 μg/mL blasticidin and 100 μg/mL hygromycin B while grown in DMEM supplemented with 10% tetracycline-free FBS until the

formation of cells grown in isolated colonies[50]. Single colonies were isolated using cloning glass cylinders (Sigma-Aldrich, C1059) and the cells were further expanded. We confirmed the presence of Dynamin1(K44A)-GFP after inducing the transgene expression with 1 μg/mL doxycycline hyclate for 18 h. We observed the presence of diffraction-limited GFP puncta across the plasma membrane of 100% of the cells using TIRF, as shown before[51]. After selection, Dynamin1-K44A-GFP Trex HEK293 cells were maintained in EMEM with 10% tetracycline-free FBS and 5 μg/mL blasticidin. For experiments, induction was performed with 100 ng/mL doxycycline hyclate overnight (western blot, Fig. S11c). All cells were maintained in sterile conditions at 37 °C in 5% $CO_2$. All HEK293 and HSC-3 cell lines were confirmed mycoplasma free and ATCC-authenticated.

### Growing HSC-3 cells on EM grids
EM grids (R2/1 Au 300 for PREM with no carbon; Au 300 R2/2, 2 nm carbon for cryoET and PREM with carbon) were first plasma cleaned for 30 s using a PELCO easiGlow (Ted Pella Inc.) and then secured to glass coverslips (1 grid/coverslip) using PDMS stencils (Nanoscale Labs 8052901) with the carbon-coated side facing up and placed into a 35 mm dish. EM grids were placed into the cell culture hood for subsequent steps starting with 20 min of UV light sterilization. Next, grids were coated with fibronectin (diluted with autoclaved $H_2O$ to reach a final concentration of 125 ng/μL, Sigma-Aldrich, F1141) for 20 min. Coated grids were washed with sterile $H_2O$ before cell seeding with an average of 15k cells (contained in a 20 μL droplet) per EM grid. Grids were transferred to an incubator at 37 °C with 5% $CO_2$ for 1 h to allow the cells to settle. After 1 h, 4 mL growth media was added to each dish, and the dishes were returned to the incubator for overnight incubation.

### Growing cells on coverslips
HSC-3 cells were seeded onto 25 mm diameter rat tail collagen I-coated coverslips (Neuvitro Corporation, GG-25-1.5-collagen) with an average of 400k cells per coverslip. Cells were then incubated overnight at 37 °C with 5% $CO_2$. HEK293 cells were seeded onto Fibronectin-coated coverslips (NeuVitro Corporation, GG-25-1.5-Fibronectin) and grown overnight before use.

### HSC-3 serum-starving
HSC-3 cells used for PREM and basal cryoET were serum-starved prior to unroofing. After overnight incubation, cells were washed with PBS at 37 °C and incubated in starvation buffer (DMEM supplemented with 1% v/v Glutamax and 10 mM HEPES) at 37 °C for 1 h followed by incubation in starvation buffer supplemented with 0.1% w/v BSA at 4 °C for 40 min.

### Cell unroofing
For generating isolated basal plasma membranes, an EM grid with cells was first washed in PBS at room temperature and then placed under an air pressure-driven fluid delivery device (Fig. S1) (ALA Scientific Instruments Inc.). The unroofing buffer (2% paraformaldehyde in stabilization buffer [30 mM HEPES, 70 mM KCl, 5 mM $MgCl_2$, 3 mM EGTA, pH 7.4]) is applied to the EM grid at between 0.7 and 0.8 bar above atmospheric pressure for 1–2 s (Fig. S2). Next, the EM grid is either washed with stabilization buffer and placed into a Leica Microsystems plunge freezer for vitrification using liquid ethane at −180 °C or placed into 2% PFA for subsequent processing for generating platinum replica.

To prepare for isolating apical plasma membranes, EM grids (Quantifoil R2/1 Au 300, Q3100AR1 for PREM, per Table 1 for cryoET) were plasma cleaned for 30 s and coated with 0.01% poly-L-lysine (Sigma-Aldrich, P4832) for 20–60 min. Cells grown on coverslips were washed with stabilization buffer and placed in stabilization buffer during the transfer of cells from a coverslip to an EM grid. Coated EM

grids were washed with $H_2O$ or stabilization buffer and then brought into contact with cells on a coverslip for 3–5 s to pick up cells off the surface. The cells on the EM grid were then unroofed and processed as described above.

## Vitrification

EM grids were washed in stabilization buffer and lightly blotted from the side, prior to adding 4 µL of fiducial solution (10 nm BSA gold tracer, Electron Microscopy Sciences, 25486; 1:1 with stabilization buffer for grids). The fiducial solution also included 500 nm carboxylate-modified red fluoSpheres (Invitrogen F8812) for grids 5 (50 µg/mL), grids 3,4,6 (25 µg/mL). Plunge freezing was performed on a Leica GP plunge freezer. The chamber was conditioned to 25 °C at 80% humidity for grids #1-2. The chamber was conditioned to 15 °C and 95% humidity for all other grids. Grids were blotted from behind for 3 s and then plunged into liquid ethane at −180 °C. Vitrified grids were transferred to liquid nitrogen for storage.

## Platinum replica electron microscopy

The generation of platinum replicas on an EM grid followed the steps as previously described[17] with slight modifications. Unroofed cells on EM grids were placed into 2% paraformaldehyde for 20 min and then transferred into 2% glutaraldehyde at 4 °C overnight. Following overnight fixation, EM grids were incubated in 0.1% tannic acid (in $H_2O$) for 20 min at 4 °C, washed with 4 °C ddH2O, and then incubated in 0.1% uranyl acetate (in $H_2O$) for 20 min at 4 °C. Grids were washed extensively with ddH2O post uranyl acetate and underwent dehydration stepwise in ethanol (15%, 30%, 50%, 70%, 80%, 90%, 100% x3) at room temperature using 4 minutes per step. EM grids were then critical point dried using an Autosamdri-815 (Tousimis). Grids were then transferred to a Leica EM ACE 900 for coating with a platinum thickness target of 3 nm (17°, 40 rpm, 110 W) and a backing carbon thickness target of 2.5 nm (90°, 40 rpm, 150 W). Coated EM grids were imaged directly (biological material still present under metal coat) using a 120 kV FEI Tecnai T12 equipped with a Rio 9 CMOS camera (Gatan). Montages were collected at 5.87 Å/pixel resolution using SerialEM[52]. Montage blending was done using the IMOD software suite[53]. We evaluated plasma membrane-grid surface interaction by segmenting and quantifying different classes of clathrin structures. Segmentation of three classes of clathrin structures—flat (no visible curvature), domed (curved lattice with a visible edge), and sphere (curved beyond a hemisphere and no visible lattice edge)—was carried out using the deep learning model Mask R-CNN[54,55]. Model training was done on 113 manually generated segmentation masks[10]. Output segmentation masks were manually reviewed, using custom built widgets for the image analysis platform Napari[56]. Masks of the whole plasma membrane area segmented, holes in the Quantifoil, and each class of clathrin structure were combined in FIJI[57]. The composite masks were then processed with MATLAB to analyze the density, projected area, and spatial organization of clathrin structures in relation to the edge of each hole in the Quantifoil carbon film for each class of clathrin structures. Projected areas were collected by counting pixels per each mask. The densities were calculated by counting the number of each type of clathrin structure mask in a given area analyzed. To evaluate the spatial organization of clathrin structures with respect to the edge of a Quantifoil hole, we first assign a centroid to the shape of each clathrin structure mask. Concentric rings, each 50 nm thick, were used to analyze a range from 500 nm inside and outside of the hole edge (20 rings; 10 inside, 10 outside). Membrane area occupation was defined as the percentage of pixels in the 50 nm-thick ring that were occupied by the segmented clathrin mask from that particular class. Density was defined as the number of structures per area analyzed. Projected area was the area of each clathrin structure in the segmented masks. Results from MATLAB were output to an Excel file for subsequent data

organization. Statistical comparisons were performed using the Mann-Whitney test (one-sided). A $p$-value of <0.05 was considered statistically significant. Data visualization and statistical tests were done using Prism (GraphPad). Platinum replica experiments were performed on the following grids: Quantifoil R1/2 Au 300 mesh ($N = 2$, basal; $N = 2$, apical), R2/1 Au 300 mesh ($N = 2$, basal; $N = 2$, apical), and on R2/2 Au 300 mesh with 2 nm C ($N = 6$, basal). Quantification was performed on R2/1 (Fig. 2) and R2/2 (Supplementary Fig. S5) grids to keep hole size consistent. For basal membranes on Quantifoil, analysis was performed on $N = 16$ membranes, 203 flat, 237 domed, 404 spherical clathrin structures from 2 R2/1 grids. For apical membranes, analysis was performed on $N = 16$ membranes, 139 flat, 206 domed, 173 spherical clathrin structures from 1 R2/1 grid. For basal membranes on Quantifoil with 2 nm carbon (Supplementary Fig. S5), analysis was performed on $N = 10$ membranes, 226 flat, 184 domed, 273 spherical structures from 1 R2/2 grid.

## Cryo-fluorescence microscopy

Cryo-fluorescence imaging was performed on a CryoCLEM Thunder Imager (Leica Microsystems) where indicated in Table 1 (Grids 3-10). Full grids were imaged with a GFP filter cube (Exc:470/40, dichroic 495, Em:525/50 Leica #11504164), a Y3 filter cube (Exc:545/25, dichroic 565, Em:605/70 Leica #11504169), and a reflector cube (BF-LP425, Leica #11505287). Final images used were single-slice extended focus projections of 35 µm thick (1 µm increments) montaged stacks (130 nm/px). Registration of the fluorescent data onto the corresponding EM atlas allowed affine transformation and was accomplished within MATLAB to target areas of interest for tomogram acquisition.

## Cryo-electron microscopy

Vitrified unroofed samples were screened by acquiring low magnification grid atlases on a 200 kV Thermo Fisher Glacios. Electron tomography was carried out on one of three 300 kV Thermo Fisher Titan Krios as indicated in Table 1. Low-magnification grid maps were acquired at 135× magnification. After coarse fluorescence image registration, grid square maps were acquired at 2250× magnification. Fluorescence image registration was finely adjusted to new grid square maps to identify tomogram positions of interest. Tilt series were collected dose-symmetrically at 42,000× magnification with a grouping of 3 using SerialEM[52] at a resolution of 1.08 Å/pixel (K3 at super-resolution mode) and 4-5 subframes per tilt from −50° to +50° at 2° increment and an evenly distributed dose targeted at 120 electrons/Å².

## Tomogram reconstruction

Tomograms were reconstructed for general purpose visual analysis separately from how they were reconstructed for STA. For general purposes, subframes were aligned at bin 8 with IMOD[58] "alignframes". For STA, subframes were aligned at bin 1. Tomogram reconstruction was performed using IMOD batch processing "batchruntomo". For general purpose, the batchruntomo directives included automatic X-ray removal, no additional binning (maintaining the bin 8 from the frame alignment), used 10 fiducials for global (no local) alignment, fiducial erasing with noise using 12 px extra diameter, and a back-projection tomogram reconstruction with a SIRT-like filter equivalent to 6 iterations. The batchruntomo directives included automatic X-ray removal, a target of 100 fiducials for global (no local) alignment, binning by 10 after stack alignment (resulting in bin10 final tomograms, 10.8 Å/px), fiducial erasing with noise using 4 px extra diameter and three expanded circle iterations, and a back-projection tomogram reconstruction with a SIRT-like filter equivalent to 10 iterations. The CTF was estimated per-tilt in IMOD, then processed with IsoNet[59] including CTF deconvolution, denoising, and modeling the missing wedge to improve the performance of automated particle picking in EMAN2[60,61].

## Sample thickness evaluation

Using the general purpose tomogram reconstructions at bin 8, each tomogram was sampled at nine equally spaced XY positions throughout the tomogram. At each location, a region of 200 × 200 voxels in XY was cropped out, Gaussian-smoothed, and observed with a minimum intensity projection along the Y-axis in the XZ plane. Using this view, the thickness was the manually determined distance between the two AWIs. For each tomogram, the average of these nine values was reported as the tomogram thickness.

## FerriTag analysis

FerriTag was automatically segmented using a trained convolutional neural network in EMAN2[61]. The output was thresholded at a 1.4 confidence level to create a 3D mask. A watershed function was used to separate nearby tags. Regions with fewer than 100 voxels were removed from the mask. The centroid of each region was used as the FerriTag position. Tags that were within 25 nm of the AWIs and tags within 75 nm of the tomogram XY edge were not used for analysis. The AWI was determined with manual segmentation and interpolation. The membrane under clathrin coats was manually segmented in IMOD and converted to 3D masks using the imodmop function. The nearest distance between each FerriTag position and the resulting clathrin-coated membrane masks was determined using the bwdist function in Matlab.

## CryoET segmentation

For Figs. 3d,e and 6g, actin, ribosomes, membrane, intermediate filaments, and FerriTag were all initially segmented with EMAN2 to create a 3D mask. These masks were manually corrected as needed to remove errant identifications. For clathrin, manual segmentation and masking were performed (IMOD, ImageJ). The masks were used to isolate signal within the tomogram for isosurface rendering. The final displayed segmentations are mask-guided isosurface renderings. Membrain[62] was used to generate the membrane segmentation in Fig. S10b.

## Subtomogram averaging and classification

Particle positions for ribosome reconstruction were imported into Dynamo[63] and subtomograms were extracted at a binning factor of 10 to generate an initial reference map. Pseudo-subtomograms were extracted and refined at a binning factor of 8 (1.0825 Å/px unbinned, 8.66 Å/px binned) followed by two rounds of classification with a soft spherical 320 Å diameter mask where junk particles were removed. The remaining 43,996 particles were visualized using the ArtiaX[64] plugin for ChimeraX[65] in each tomogram. A population of ribosomes was located at the underside of the carbon film, possibly due to positively charged poly-L-lysine-coated carbon film attracting negatively-charged ribosomes from the cytoplasm during unroofing. These film-adhered ribosomes were discarded from analysis. Refinement in RELION[66] was performed on the remaining 23,619 particles at a binning factor of 4 using a soft mask around the entire complex. Further, refinement was performed at binning 2 with a mask focusing on the LSU, followed by classification without alignment ($K = 10$). One class, containing 11,288 particles, was chosen, based on the rlnAccuracyRotations, rlnAccuracyTranslationsAngst, and rlnEstimatedResolution parameters. Structures of non-rotated and rotated ribosomes were obtained by classification ($K = 2$) without alignment of the consensus particle set using a mask focused on the SSU. In a separate classification ($K = 12$), a single class of membrane-bound ribosomes (446 particles) was obtained from the consensus particle set without alignment using a mask focusing on the peptide exit tunnel and the surrounding region (Fig. S9). Three refinement cycles of CTF and frame alignment parameters with refinement of the consensus particle set were performed. Final maps were obtained after unbinned refinement. Maps were locally filtered and sharpened in RELION 4.0. Clathrin vertices were manually picked in 20

tomograms from grid 6 reconstructed at 10 Å/px. 2435 particles were initially aligned to a reference structure (EMD-0122) at a binning factor of 6 (1.058 Å/px unbinned, 6.348 Å/px binned). One round of classification was performed, identifying one well-aligning class of 1,550 particles, which was then refined with C3 symmetry at a binning factor of 2. Each of these steps was performed in RELION 5.0[67] for the clathrin data.

## Ribosome distribution analysis

Tomograms processed with IsoNet, at 10.825 Å/px, were examined in 3dmod[53] and points were manually defined to model the top and bottom AWIs in tomograms of unroofed cells. The top and bottom interfaces were determined with model points placed based on visual inspection of the tomograms, fiducial positions, and ice. A custom Python script was used to interpolate the picked model points for each interface and calculate distances from each particle to the interpolated surfaces.

## Live TIRF microscopy

HEK293 cells transfected with *FTL*, *FRB-mCherry-FTH1*, and *HIP1R-GFP-FKBP* were grown on fibronectin-coated coverslips and imaged with total internal reflection fluorescence (TIRF) microscopy using Nikon Eclipse TI inverted fluorescence microscope with a 100× apoTIRF 1.49 NA objective and 488 nm and 561 nm excitation lasers. Each color was imaged with 100 ms exposure every 30 s. The sample was kept in growth media at 37 °C and 5% $CO_2$ during imaging. 200 nM rapamycin was added between frames of the movie.

## Reporting summary

Further information on research design is available in the Nature Portfolio Reporting Summary linked to this article.

## Data availability

The data that support this study are available from the corresponding authors upon request. Platinum replica data are available on Figshare [https://doi.org/10.25444/nhlbi.28012229]. Binned tomograms and raw frames are available on the Chan Zuckerberg Initiative CryoET Data Portal under accession code CZCDP-10306. Ribosome structures are being made available as EMD-44921, EMD-44909, and EMD-44922. The clathrin structure is being made available as EMD-46973. Atomic coordinates of previously determined X-ray or Cryo-EM structures are available in the PDB under the following accession codes: 6YAI (clathrin with bound β2 appendage of AP2), 1FHA (Ferritin), 2H5Q (mCherry), 3FAP (RB/FKBP), 5WWK (GFP). Source data are provided with this paper. The source data underlying Figs. 2f–i, 4c & h, 6f, S3b–d, S5d, and S8 are provided as a Source Data file.

## Code availability

Code used for subtomogram averaging is available at https://github.com/dmichalak/sta-pipeline. Code for image correlation is available at https://github.com/KASochacki/clemposo. Code for clathrin segmentation in platinum replica data is available at https://github.com/andreasmarnold/PyCLEM and https://zenodo.org/records/14391734.

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

## Acknowledgements

We thank the NIH Intramural CryoEM Consortium (NICE), and the NIH Multi-Institute Cryo-Electron Microscopy Facility (MICEF) for use of equipment, data acquisition support, and data management support; specifically, Rick Huang of NICE, and Huaibin Wang, Bertram Canagarajah, and Ulrich Baxa of MICEF. We thank Alexander Sorkin (University of Pittsburgh, USA) for the generous gift of the HSC-3/EGFR-GFP cell line; Jiamin Liu of the NIH Advanced Imaging and Microscopy Resource (AIM) for help training mask R-CNN for platinum replica segmentation; NIH HPC Biowulf cluster for computational resources; Naoko Mizuno for discussions during project initiation; Ethan Tyler of NIH Medical Arts for creating Fig. 1. This project was made possible in part by grant 2021-234544 from the Chan Zuckerberg Initiative DAF, an advised fund of the Silicon Valley Community Foundation. JWT is supported by the Intramural Research Program (IRP) of NHLBI, NIH through 1ZIAHL006098. J.E.H. is supported by the IRP of NIDDK, NIH through 1ZIADK060100.

## Author contributions

W.W.S., D.J.M., and K.A.S. functioned as the core team to acquire, discuss, analyze data, and write the manuscript. K.A.S. was team leader. W.W.S. performed/analyzed unroofing tests, PREM, cryoET grids 1–2. D.J.M. performed unroofing tests and STA. K.A.S. performed/analyzed cryoET grids 3–10. K.A.S. and M.-P.S. performed western blots. P.K. segmented tomograms. M.A.A. and M.-P.S. performed/supported plasmid/cell line preparations. A.M.A. provided Python script support for PREM segmentation. K.A.S., J.W.T., and J.E.H. guided experiments with regular feedback. W.W.S., D.J.M., K.A.S., J.E.H., and J.W.T. edited manuscript.

## Funding

## Competing interests

The authors declare no competing interests.
