## [Transparent Peer Review file · Nature Communications]

Cryo-electron tomography pipeline for plasma membranes.

Corresponding Author: Dr Kem Sochacki

Version 0:

Reviewer comments:

Reviewer #1

(Remarks to the Author)

The authors developed a unique correlative light and electron microscopy (CLEM) protocol to examine protein interactions at apical or basal plasma membranes in mammalian cells. This pipeline includes “cell unroofing,” a technique that allows either the apical or basal membrane of a cell to remain intact and attached to an EM grid while retaining membrane-associated protein complexes, including clathrin-associated proteins. This protocol produced samples between 78nm-221nm, rivaling that of FIB-milled lamella, producing ideal samples for cryo-electron tomography. These samples also display comparative ribosomal structures to control cells, suggesting that the protein structure and integrity are intact after the cellular unroofing protocol, helping to support this protocol as a new and ideal method for visualizing membrane-bound proteins. The authors could also clearly show a variety of clathrin-coated vesicles associating with cell membranes, suggesting that this method is sufficient for visualizing clathrin-mediated endocytosis and its associated proteins through this method. While this protocol produced cryo-ET data that allowed structures to be visualized, the authors found that an ideal tag for CLEM was iron-free Ferrin, allowing for proper visualization and correlation of structures through cryo-ET. Overall, the authors present an interesting and unique alternative to conventional FIB-milling to maintain proper protein-protein interactions and protein-membrane bound interactions for novel membrane protein structure determination. While the work is interesting and provides a novel protocol for examining membrane-protein interactions, the bulk focuses on this protocol establishment without new findings or mechanisms. With this in mind, this reviewer questions whether this publication is the best fit for Nature Communications and should be considered for acceptance with revisions being performed.

Major Concerns:

1. The authors claim to use cell lines stably expressing Dynamin(K44A)-GFP or transiently expressing either GFP-FKBP-LCa/FerriTag or Hip1R-GFP-FKBP/FerriTag but do not provide any western blot analysis showing their expression within cells. Western blots must be performed and provided to show the maintenance of stable cell lines and successful transfections.
2. The authors do not provide novel findings within this manuscript. The discussion even states that Figure 6 confirms the data previously published (Sochacki et al. 2017, Nat Cell Bio) and that this new data only shows this result at a higher resolution. While the significance of this resolution increase should not be ignored, this data does not seem to be enough to warrant publication within Nature Communications since it lacks novel implications or a new mechanistic understanding of membrane proteins and their interactions.

Minor Concerns:

1. Figure 1: The unroofing protocol lacks some evidence for repetition and parameter establishment. It is unclear where the authors came up with the pressure values for the unroofing and the lack of real-time photos that correspond to with the diagrams in Figure 1 just leave me questioning how this protocol was thoroughly tested. Can the authors provide more step-by-step photographs to support this workflow? Additionally, were other concentrations of compressed air used to test optimal sheering conditions or to see if these conditions need to be modified for different cell lines? This display of rigor will be advantageous in supporting the authors claim that this is a beneficial new protocol for others to utilize.
2. Figure 2/Figure S4: The graphs in these figures are a bit cumbersome. The authors should either overlay the individual datapoints with the box-and-whisker plots or just show one or the other since it is not initially clear that the two columns are representing the same datasets.
3. In line #86, where the authors state “flat and sphere-shaped 85 clathrin structures found in the basal membranes were larger (in μm^2 : flat, 0.032 ± 0.022 ; dome, $0.019 \pm 86 \ 0.008$; sphere, 0.013 ± 0.003) than their apical counterparts (in μm^2 : flat,

0.023 ± 0.012; dome, 0.019 ± 87 0.008; sphere, 0.011 ± 0.004)." However, the differences in size are within the error range and hence does not seem to be significant.

4. In lines #122-124, "A peak in the particle distribution, indicating accumulation, occurred at 25-30 nm from the top AWI (Fig. 4g, asterisk). In contrast, the distribution measured from the bottom AWI gradually increased and plateaued at a distance of approximately 80 nm." The asterisk mark mentioned seems to be marked at different position than intended. Also, the distribution measured from the bottom AWI plateaus before 80 nm, authors should correct this. The authors do explain the asterisk and dashed lines used in the plots in Figure 4, but I would recommend mentioning those in figure caption too, makes it easier to follow the figure.

5. In Figure 5f and 5g, the authors should mark Dynamin 1 and clathrin densities for the general audience to follow.

6. The authors report large arrested clathrin structures that they refer to as clathrin grape clusters. Although these sites had fluorescence, the authors could not identify dynamin 1 at these sites, which the authors presume could be due to dynamin assemblies harder to distinguish. Using the FerriTags to identify dynamin 1 at these sites would be an excellent use of the tool, which would provide novel information about dynamin 1 assembly and distribution at these sites.

Reviewer #2

(Remarks to the Author)

In this manuscript entitled "Cryo-electron tomography pipeline for plasma membranes", Sun et al. from the Taraska lab present a correlative cryo-electron tomography pipeline to image large ultra-thin areas of isolated basal or apical plasma membranes for correlative cryo-ET. They found differences in macromolecular assemblies such as the different degrees of clathrin-coated pit curvature between the apical and basal membranes and show that, when cells are cultured on EM grids, membrane-associated protein complexes adapt to the grid surface topography and that cell unroofing preserves sub-nanometer resolution. To locate rare proteins or proteins of unknown structure, they implemented a CLEM protocol. First, using the well-characterized dominant negative mutant of dynamin 1, Dyn1(K44A)-GFP, that blocks clathrin-mediated endocytosis, they were able to find clathrin-coated pits forming grapes and longer tubes with putative dynamin helices around them although unfortunately dynamin could not be identified in the tomograms. They then use correlative light and electron microscopy (CLEM) coupled to iron free FerriTag technique used here as a chemically-inducible tag to locate and observe clathrin associated proteins such as Hip1R and distinguish the molecular shape of Hip1R coupled to ferritin structures in their micrographs.

Overall, the authors have generated a new optimized pipeline for cryoET of plasma membranes capable of identifying proteins for structural cellular biology. This is a very interesting and elegant study showing how Cryo-ET can be used to analyse unroofed membranes at the highest possible resolution. The imaging is of the highest quality and the manuscript is very well written. There are still however some points that will need to be improved/considered for the manuscript to be more coherent as a whole:

Major comments:

1/ In order to show that high-resolution structural information is preserved in unroofed cells, the authors use ribosomes which are abundant in their preparations (Fig 4). Although understandable as ribosomes are very abundant in such preparations, these experiments are taking the reader away from the main interest which is centred on clathrin-coated pits. Why didn't the authors perform the same analysis on clathrin-coated pits at the surface which are also abundant, fully characterized by cryo-EM (<https://www.rcsb.org/structure/1xi4>) and in the scope of this manuscript?

2/ In Fig 2, the density of the different clathrin pits at different degrees of curvature (flat-dome-sphere) is shown on cells cultured on EM grids. How do these distributions compare to cells grown on glass? These experiments have all been performed on HSC tongue squamous carcinoma cell line. The choice of cell type is not discussed but is rather unusual. How do these cells compare to more conventional or more physiological cell types? are these the same in MEFs or the more frequently used HeLa cells?

3/ In Fig. 2 the authors show images corresponding to the basal and apical membrane at low magnification but never show the ultrastructure of the apical membrane. Similarly, in Fig 3, while both the basal and apical membranes are shown at high mag, the panel corresponding to the apical side (Fig 3 panel e) does not display any coated pit. Since the authors make a point at quantifying the clathrin structures, representative clathrin pits from the apical side should also be shown in both PREM and Cryo-EM and maybe highlighted with an arrow for those who are less familiar with these EM techniques.

4/ In the same line as the point raised above, the authors attempted to visualize dynamin spirals in Dyn1 K44A expressing cells at sites of arrested endocytosis. However, in these experiments Dyn1 could not be identified in the tomograms. This is very unfortunate as it would provide indeed compelling evidence for the usefulness of this technique but also provide stronger rationale for this work. The authors propose that this is presumably because Dyn1 is assembled in short spirals or other assemblies that are indistinguishable. I'm a bit puzzled by this explanation as these assemblies are clearly visible on their cryo-EM examples of arrested CME sites either as spirals at the neck of arrested pits or as longer decorated tubules. This part should be strengthened in the revised manuscript. Can the authors show the sub-tomogram averages corresponding to the densities observed in the Dyn1 K44A mutant expressing cells?

5/In Fig. 6 the authors show putative molecules and claim that it is Hip1R. Although this is an interesting finding, the authors should provide more convincing evidence that these are indeed Hip1R molecules. Alternative labelling approaches such as immunogold labelling or correlative EM with a higher resolution light microscopy could convincingly prove that these are indeed bona fide Hip1R molecules. Maybe it would be beneficial to show a gallery with more than two examples of Hip1R molecules bound to the Ferritag. Can the authors go back to their PREM replicas and now distinguish Hip1R?

6/Also in Fig. 6, in panels g and h the authors show only small insets. It is not clear if they come from panel (g), however the whole image should be shown and some way to quantify the amount of Hip1R bound to ferritin should be performed, otherwise, these are only anecdotal findings that would be otherwise difficult to reproduce.

7/The Hip1R molecules seen in Fig 6 are depicted with a cartoon next to the EM inset. Hip1R is shown to bind both a ferritin molecule and an actin filament. How can the authors be sure this is actin? although actin can be identified without labeling, it is impossible for the reader to tell as only a small portion of the filament is shown.

Version 1:

Reviewer comments:

Reviewer #2

(Remarks to the Author)

The authors have satisfactorily addressed all the points that were raised during my initial screening by either performing additional experiments or quantifying them, by adding additional images and rewording the main text. I am happy to see this work published in this journal as it brings valuable new techniques bridging cryo-EM and PREM and provides experimental evidence of Hip1R ultrastructure and localization around clathrin coated pits.

Sun, Michalak, Sochacki et al. Nature Communications. Response to reviewers' comments:

We want to thank the reviewers and editor for their time and effort in providing constructive feedback on our manuscript. To address the questions and concerns raised by the reviewers, we have conducted new experiments, extended our analysis, determined new structures, and revised the text & figures accordingly. Five additional supplementary figures have been added to strengthen the rigor, impact, and depth of this manuscript. A detailed response to each comment is provided below. The green text in the manuscript are major sections that have been edited. We hope the manuscript is now suitable for publication. Thank you again.

REVIEWER COMMENTS

Reviewer #1 (Remarks to the Author):

The authors developed a unique correlative light and electron microscopy (CLEM) protocol to examine protein interactions at apical or basal plasma membranes in mammalian cells. This pipeline includes “cell unroofing,” a technique that allows either the apical or basal membrane of a cell to remain intact and attached to an EM grid while retaining membrane-associated protein complexes, including clathrin-associated proteins. This protocol produced samples between 78nm-221nm, rivaling that of FIB-milled lamella, producing ideal samples for cryo-electron tomography. These samples also display comparative ribosomal structures to control cells, suggesting that the protein structure and integrity are intact after the cellular unroofing protocol, helping to support this protocol as a new and ideal method for visualizing membrane-bound proteins. The authors could also clearly show a variety of clathrin-coated vesicles associating with cell membranes, suggesting that this method is sufficient for visualizing clathrin-mediated endocytosis and its associated proteins through this method. While this protocol produced cryo-ET data that allowed structures to be visualized, the authors found that an ideal tag for CLEM was iron-free Ferrin, allowing for proper visualization and correlation of structures through cryo-ET. Overall, the authors present an interesting and unique alternative to conventional FIB-milling to maintain proper protein-protein interactions and protein-membrane bound interactions for novel membrane protein structure determination. While the work is interesting and provides a novel protocol for examining membrane-protein interactions, the bulk focuses on this protocol establishment without new findings or mechanisms. With this in mind, this reviewer questions whether this publication is the best fit for Nature Communications and should be considered for acceptance with revisions being performed.

Major Concerns:

1. The authors claim to use cell lines stably expressing Dynamin(K44A)-GFP or transiently expressing either GFP-FKBP-LCa/FerriTag or Hip1R-GFP-FKBP/FerriTag but do not provide any western blot analysis showing their expression within cells. Western blots must be performed and provided to show the maintenance of stable cell lines and successful transfections.

Thank you for this comment. In correlative light and electron microscopy, we have the benefit of observing expression of these GFP-containing constructs directly at a single cell level with fluorescence. In the experiments involving the constructs discussed above (Dynamin1(K44A)-GFP and FerriTag) correlative fluorescence microscopy was always performed (indicated in Table 1) to confirm expression of the protein in the imaged cell and to zero in on specific sub-cellular locations for EM. The localization of the fused fluorescent proteins to endocytic clathrin sites further indicates that the expressed proteins are correctly targeted to their biological sites of action. There is one component of FerriTag (Ferritin light chain, FTL) that is not fluorescent. To further show that this protein is behaving correctly, we have now confirmed expression with Western blots as suggested by the reviewer (Fig. S11b). We also confirmed expression of Ferritin heavy chain (FRB-mCherry-FTH1; Fig. S11a) and Dynamin (K44A)-GFP in our stable Trex cell line (Fig. S11c). We hope these new control data strengthen the rigor of our study.

2. The authors do not provide novel findings within this manuscript. The discussion even states that Figure 6 confirms the data previously published (Sochacki et al. 2017, Nat Cell Bio) and that this new data only shows this result at a higher resolution. While the significance of this resolution increase should not be ignored, this data does not seem to be enough to warrant publication within Nature Communications since it lacks novel implications or a new mechanistic understanding of membrane proteins and their interactions.

This concern is appreciated. We do feel that there are several novel findings in our manuscript. As this is a methods paper, however, our primary motivation is to present the method in a clear, concise, and reproducible manner so that it can become a useful and rapidly distributed tool for the structural cell biology community. To better showcase the novel findings in the paper, we have now enhanced three specific areas to highlight these discoveries. We discuss them below.

- 1) The work done here is a significant resolution improvement over our previous data on Hip1R with platinum replica electron microscopy (PREM) super-resolution fluorescence CLEM (Sochacki et al. Nat Cell Bio 2014). However, the techniques are quite different and have very different strengths. In this regard, speaking only in terms of resolution is not necessarily the best comparison between these two studies. With cryoET, we can see the exact position of FerriTag (with no image registration error) in 3-dimensions (compared to PREM 2D measurements of the surface) and observe the actual carbon-based atomic density of proteins near the probe in the tomogram (rather than a 2-3 nm metal cast of the surface). We have re-worded the discussion to better describe the way this work is

superior to our previous work and provides a completely different view of this protein in the context of endocytosis.

- 2) Our previous version of the manuscript did not thoroughly describe how our new Hip1R data fit into previous models of the protein's structure and function. The discussion paragraph describing Hip1R now gives more background, pointing to areas where our data answer open questions in the field.

“Hip1R/actin binding has previously been shown to be regulated by clathrin light chain. The theory postulates that Hip1R present throughout the clathrin lattice would not bind actin as readily as the Hip1R outside of the clathrin lattice at the budding membrane neck and may bend or fold inward to inhibit actin binding. Here, we find a dense network of actin and Hip1R-bound FerriTag uniformly surrounding the clathrin structures consistent with Hip1R actively binding actin throughout the clathrin lattice. While our data are consistent with Hip1R bending, it is in close proximity to actin. FerriTags line up along actin fibers (Fig. 6g) emphasizing the need for some requisite Hip1R bending to connect a linear fiber to a round clathrin structure.”

- 3) Finally, we have added an entirely new section of subtomogram averaging of the clathrin vertex in lattices assembled at the plasma membrane of a living cell. Here, we present new evidence that clathrin curvature is accommodated by splaying of the heavy chain N-terminus. This structural change is important because it is the region in which the major adaptor protein complex AP-2 binds the clathrin lattice. Thus, a change in lattice curvature could potentially create a change in adaptor binding or vice versa. These are new models of how clathrin functions. Extensive future work will be needed to refine these structures and provide detailed atomic models. We are excited about this direction for the method and our research questions into the structure of the clathrin lattice.

We appreciate these comments and hope that our changes help to highlight not only the novelty of our biological observations along with the potential of the method to understand the structure of the plasma membrane and its associated proteins and organelles. This has been a gap in the structural biology of the cell.

Minor Concerns:

1. Figure 1: The unroofing protocol lacks some evidence for repetition and parameter establishment. It is unclear where the authors came up with the pressure values for the unroofing and the lack of real-time photos that correspond to with the diagrams in Figure 1 just leave me questioning how this protocol was thoroughly tested. Can the authors provide more step-by-step photographs to support this workflow? Additionally, were other concentrations of

compressed air used to test optimal sheering conditions or to see if these conditions need to be modified for different cell lines? This display of rigor will be advantageous in supporting the authors claim that this is a beneficial new protocol for others to utilize.

We love the idea of including step-by-step photos so readers can better follow the sample preparation process. Thanks for the suggestion. We have included real-life photos (in Fig. S2) that correspond to the cartoon depictions (in Fig. 1).

We now also include a new parameter screen across pressure and distance for unroofing (Fig. S3). As noted, different pressures and different cell lines are major factors that could affect unroofing success. We conducted a series of new experiments to show how the pressure of the unroofing stream affects unroofing using two cell lines that are constitutively expressing a fluorophore-tagged protein, HSC3-EGFR-GFP and MDA-MB231-CLCa-GFP. We quantified the effective radius of unroofing at different syringe pressures and different distances between the sample and the syringe (the distance travelled by the fluid would affect the pressure the sample experiences). Our data indicate that pressure is likely the dominant parameter that determines the outcome of cell unroofing in an individual cell line with our device. While the parameters used throughout the manuscript work well with all cell lines we tested, different cell lines unroof differently. This is now shown in the new parameter screen. We also included a cryo-fluorescence image of cell unroofing using the optimal parameters described in the manuscript to show an overview of the whole grid after unroofing. Please see Fig. S3 and the supplementary method for additional details. We hope these additions improved the usefulness of our manuscript.

2. Figure 2/Figure S4: The graphs in these figures are a bit cumbersome. The authors should either overlay the individual datapoints with the box-and-whisker plots or just show one or the other since it is not initially clear that the two columns are representing the same datasets.

We understand the concern that these plots are cumbersome to view. We do prefer, however, to show both the box/whisker statistical plot and all the individual data points for rigor and transparency of data presentation. We have now modified the plots of concern (Fig. 2 and Fig. S8) to use color and spacing to better group the data and make it easier to read. We hope this is now acceptable.

3. In line #86, where the authors state “flat and sphere-shaped 85 clathrin structures found in the basal membranes were larger (in μm^2 : flat, 0.032 ± 0.022 ; dome, $0.019 \pm 86 0.008$; sphere, 0.013 ± 0.003) than their apical counterparts (in μm^2 : flat, 0.023 ± 0.012 ; dome, $0.019 \pm 87 0.008$; sphere, 0.011 ± 0.004).” However, the differences in size are within the error range and hence does not seem to be significant.

Thank you for pointing out that the statistical significance between groups is not obvious in the main text. p -values are now included in the main text to clearly indicate the result of the significance test so that readers can better follow the quantifications presented.

4. In lines #122-124, "A peak in the particle distribution, indicating accumulation, occurred at 25-30 nm from the top AWI (Fig. 4g, asterisk). In contrast, the distribution measured from the bottom AWI gradually increased and plateaued at a distance of approximately 80 nm." The asterisk mark mentioned seems to be marked at different position than intended. Also, the distribution measured from the bottom AWI plateaus before 80 nm, authors should correct this. The authors do explain the asterisk and dashed lines used in the plots in Figure 4, but I would recommend mentioning those in figure caption too, makes it easier to follow the figure.

Thank you for pointing this out. We agree that the asterisk was misplaced and did not help in the interpretation of the data. The asterisk and discussion of a plateau in the distribution of ribosomes has been removed.

5. In Figure 5f and 5g, the authors should mark Dynamin 1 and clathrin densities for the general audience to follow.

We have enhanced Fig. 5 to clearly highlight Dynamin 1 and clathrin densities with arrows.

6. The authors report large arrested clathrin structures that they refer to as clathrin grape clusters. Although these sites had fluorescence, the authors could not identify dynamin 1 at these sites, which the authors presume could be due to dynamin assemblies harder to distinguish. Using the FerriTags to identify dynamin 1 at these sites would be an excellent use of the tool, which would provide novel information about dynamin 1 assembly and distribution at these sites.

We agree that tagging Dynamin 1 with FerriTag would be an incredible and powerful use of the system. We also think that studying the localization and structure of dynamin (the 100 kDa GTPase that is thought to control endocytic membrane fission) with FerriTag would come with a large body of controls and conditions that are beyond the scope of this manuscript. Ongoing work in our labs is devoted to solving the structures of Dynamin inside cells. In this manuscript, significant time and effort has been put into the essential steps of optimizing the unroofing method for cryoET and CLEM, testing its utility for sub-tomogram averaging, and finding a labeling method that works well in unroofed cell samples to identify proteins of unknown structure or that are rare.

Thank you for your helpful comments. We hope the paper is now improved and acceptable.

Reviewer #2 (Remarks to the Author):

In this manuscript entitled "Cryo-electron tomography pipeline for plasma membranes", Sun et al. from the Taraska lab present a correlative cryo-electron tomography pipeline to image large ultra-thin areas of isolated basal or apical plasma membranes for correlative cryo-ET. They found differences in macromolecular assemblies such as the different degrees of clathrin-

coated pit curvature between the apical and basal membranes and show that, when cells are cultured on EM grids, membrane-associated protein complexes adapt to the grid surface topography and that cell unroofing preserves sub-nanometer resolution. To locate rare proteins or proteins of unknown structure, they implemented a CLEM protocol. First, using the well-characterized dominant negative mutant of dynamin 1, Dyn1(K44A)-GFP, that blocks clathrin-mediated endocytosis, they were able to find clathrin-coated pits forming grapes and longer tubes with putative dynamin helices around them although unfortunately dynamin could not be identified in the tomograms. They then use correlative light and electron microscopy (CLEM) coupled to iron free FerriTag technique used here as a chemically-inducible tag to locate and observe clathrin associated proteins such Hip1R and distinguish the molecular shape of Hip1R coupled to ferritin structures in their micrographs.

Overall, the authors have generated a new optimized pipeline for cryoET of plasma membranes capable of identifying proteins for structural cellular biology. This is a very interesting and elegant study showing how Cryo-ET can be used to analyse unroofed membranes at the highest possible resolution. The imaging is of the highest quality and the manuscript is very well written. There are still however some points that will need to be improved/considered for the manuscript to be more coherent as a whole:

Major comments:

1/ In order to show that high-resolution structural information is preserved in unroofed cells, the authors use ribosomes which are abundant in their preparations (Fig. 4). Although understandable as ribosomes are very abundant in such preparations, these experiments are taking the reader away from the main interest which is centred on clathrin-coated pits. Why didn't the authors perform the same analysis on clathrin-coated pits at the surface which are also abundant, fully characterized by cryo-EM (<https://www.rcsb.org/structure/1xi4>) and in the scope of this manuscript?

Thank you for this suggestion. Clathrin was not an initial target for our method because it is notoriously flexible and, thus, not an ideal target for testing and validating high-resolution structural determination in these new samples. None-the-less, our lab has had a long standing interest in clathrin structure and we now included subtomogram averaging of clathrin lattices (Fig. 5h-I, Fig. S5). Our data differ from previous high resolution in-gel particle analysis (SPA) or sub-tomogram averaging (STA) clathrin data in that our lattices are not purified but in a pre-endocytic, and lower curvature state, still attached to the plasma membrane. The resulting structure is now compared to previously-described models. Notably, the N-terminus of clathrin heavy chain is offset from previous models, which is likely due to the differences in how the

triskelia splay to accommodate different curvatures in cells. This is a region of adaptor binding (which we observe) and thus could impact the adaptor/lattice relationship during curvature. We are happy to now include these new data as a demonstration of the type of data contained within our tomograms and the potential of the method.

2/In Fig 2, the density of the different clathrin pits at different degrees of curvature (flat-dome-sphere) is shown on cells cultured on EM grids. How do these distributions compare to cells grown on glass? These experiments have all been performed on HSC tongue squamous carcinoma cell line. The choice of cell type is not discussed but is rather unusual. How do these cells compare to more conventional or more physiological cell types? are these the same in MEFs or the more frequently used HeLa cells?

Thanks for pointing this out. We agree that we could better communicate the reasoning behind using the HSC3 cell line and provide points of comparison with other cell lines. In our lab's previous publication (Alfonzo-Mendez et al. Nat. Comm. 2022) HSC3 cells were grown on coverslips and unroofed. The distribution of clathrin-coated structures in these samples have been extensively characterized. Their previous characterization (both apical and basal) on coverslips and the fact that they had a fluorescently tagged receptor (EGFR) in the membrane made this cell line particularly useful in testing and characterizing this unroofing method. We have updated our text to include a comparison of clathrin-coated structure distribution between HSC3 cells on the EM grid and glass coverslip. We have also evaluated a variety of cell lines, including more conventional lines such as 3T3 and HeLa cells in our prior publication (Sochacki et al., Dev. Cell 2021). The text has been revised to include a comparison between HSC3 cells on grids and 3T3, BS-C-1, and HeLa cells on coverslips. While this is not a direct comparison, this information helps provide a reference to compare clathrin density and shape data across commonly used cells. Please see the revised text for details.

3/In Fig. 2 the authors show images corresponding to the basal and apical membrane at low magnification but never show the ultrastructure of the apical membrane. Similarly, in Fig 3, while both the basal and apical membranes are shown at high mag, the panel corresponding to the apical side (Fig 3 panel e) does not display any coated pit. Since the authors make a point at quantifying the clathrin structures, representative clathrin pits from the apical side should also be shown in both PREM and Cryo-EM and maybe highlighted with an arrow for those who are less familiar with these EM techniques.

Thank you for bringing to our attention that the figures do not clearly showcase ultrastructural details of basal and apical membranes in PREM and in Cryo-EM. We have included new montages from PREM to provide a better visualization of isolated apical and basal plasma membranes. Please see Fig. S6 and S7. The three classes of clathrin-coated structures are color-coded in the

images. We have updated Fig. 3 to show an additional example tomographic projection of the isolated apical plasma membrane containing clathrin-coated structures.

4/In the same line as the point raised above, the authors attempted to visualize dynamin spirals in Dyn1 K44A expressing cells at sites of arrested endocytosis. However, in these experiments Dyn1 could not be identified in the tomograms. This is very unfortunate as it would provide indeed compelling evidence for the usefulness of this technique but also provide stronger rationale for this work. The authors propose that this is presumably because Dyn1 is assembled in short spirals or other assemblies that are indistinguishable. I'm a bit puzzled by this explanation as these assemblies are clearly visible on their cryo-EM examples of arrested CME sites either as spirals at the neck of arrested pits or as longer decorated tubules. This part should be strengthened in the revised manuscript. Can the authors show the sub-tomogram averages corresponding to the densities observed in the Dyn1 K44A mutant expressing cells?

Thanks for bringing this up so we could clarify our results. We do see well-defined and easily identifiable dynamin spirals in 10% of the clathrin structures in our tomograms. However, most (90%) of our arrested clathrin structures observed in the Dyn1 K44A mutants don't have long obvious dynamin spirals. The dynamin spirals that do exist have different diameters and are heterogeneous. This indicates that extremely large datasets would be needed for good high-quality subtomogram averaging of dynamin. We did try to perform subtomogram averaging (STA) on our existing dataset, but feel that the result is not helpful to distribute to the scientific community at this time. However, the enrichment of clathrin in these data are helpful in performing STA on plasma membrane-bound clathrin lattices, which has not been previously done and is now included in the manuscript. We have tried to better communicate the dynamin results and now include clathrin STA and its associated novel findings in Fig. 5.

5/In Fig. 6 the authors show putative molecules and claim that it is Hip1R. Although this is an interesting finding, the authors should provide more convincing evidence that these are indeed Hip1R molecules. Alternative labelling approaches such as immunogold labelling or correlative EM with a higher resolution light microscopy could convincingly prove that these are indeed bona fide Hip1R molecules. Maybe it would be beneficial to show a gallery with more than two examples of Hip1R molecules bound to the Ferritag. Can the authors go back to their PREM replicas and now distinguish Hip1R?

We understand your concern. We agree that more examples would much better communicate our observations of Hip1R. We have now included 6 new supplemental movies (5 Hip1R and 1 clathrin light chain) that allow the reader to see clathrin structures surrounded with many FerriTags in 3-dimensions directly. We have also added an additional zoomed out view of Fig. 6h to add better context to this figure.

Protein labeling at this high of a resolution is, indeed, a very difficult thing to confirm. Here, we have confirmed with two-color fluorescence that the tag is colocalizing with the protein of interest at the resolution of diffraction-limited light microscopy. Next, we have quantified the position of the tags with respect to clathrin coated membrane with EM. As we know these proteins are located around clathrin structures, this quantification is being used as a test of tag specificity. Beyond this, we can see that when we tag clathrin light chain, FerriTag is where we expect it to be. When we tag Hip1R, we see the tag is in a location consistent with the predicted length of Hip1R. We further see putative density that extends between the clathrin lattice and actin. The shape of Hip1R matches its predicted and *in vitro* visualized structure. While we agree that alternative labelling approaches are ideal, we do not believe that existing alternative methods would more convincingly prove that these are indeed Hip1R molecules. Immunogold, for example would place a large and dense gold particle 25 nm away from the protein of interest and obscure the protein density surrounding it (due to its opacity in EM). Super-resolution fluorescence correlative cryoET has not yet successfully identified the density of single proteins at the atomic scale due to low image registration precision, sample damage during cryo-fluorescence imaging, and low-throughput. Still, we agree that in the dense milieu of the cell, FerriTag can be bound to Hip1R and still be next to other densities that we might misidentify as Hip1R. Thus, the labelled densities in cellular cryoET are putative. We have modified the text to use this type of language out of an abundance of caution. Thank you for this comment. We hope this is now acceptable.

6/Also in Fig. 6, in panels g and h the authors show only small insets. It is not clear if they come from panel (g), however the whole image should be shown and some way to quantify the amount of Hip1R bound to ferritin should be performed, otherwise, these are only anecdotal findings that would be otherwise difficult to reproduce.

Thank you for this advice. Additional panels have been added to Fig. 6 along with supplementary movies of each image to allow the reader to directly assess these observations. The data in Fig. 6g-h are part of the quantification in Fig. 6f. As the ferritin is specifically localized to clathrin structures in cryoET and fluorescently colocalized with Hip1R only after rapamycin addition, we believe it is reasonable to conclude they are bound to the clathrin-associated protein, Hip1R.

7/The Hip1R molecules seen in Fig 6 are depicted with a cartoon next to the EM inset. Hip1R is shown to bind both a ferritin molecule and an actin filament. How can the authors be sure this is actin? although actin can be identified without labeling, it is impossible for the reader to tell as only a small portion of the filament is shown.

We have now rearranged Fig. 6 to allow for better visualization of this panel. We have also added Supplemental Movies so the reader can observe a large portion of the actin filament at many different Z-planes. Actin itself is readily identifiable in cryoET due to its characteristic width and periodicity. However, it is sometimes difficult to get all components in the same 2D

plane for displaying purposes. We hope that the 3D stacks and zoomed out views help readers to better observe the context for our assessments and the quality of our data.

Thank you for your time and effort on reviewing this manuscript. We hope the paper is now improved and acceptable for publication.